# KLF5 enables dichotomous lineage programs in pancreatic cancer via the AAA+ ATPase coactivators RUVBL1 and RUVBL2

Patrick J. Cunniff [1,2], Nicole Sivetz[1,2], Damianos Skopelitis[1], Olaf Klingbeil[1], Daniel Toobian[1,3], Diogo Maia-Silva[1,2], Mikala Egeblad [4] & Christopher R. Vakoc [1] ✉

Lineage plasticity is a hallmark of pancreatic ductal adenocarcinoma (PDAC) and contributes to tumor heterogeneity and therapeutic resistance. Here, we identify KLF5 as a dynamic master regulator of epithelial lineage identity in PDAC, with dichotomous roles in promoting either classical or basal-like transcriptional programs. Through unbiased proteomic and genetic screens, we uncover the AAA+ ATPases RUVBL1 and RUVBL2 as essential coactivators of KLF5 across both lineage states. We demonstrate that ATP hydrolysis by RUVBL1/2 is required for the stable interaction with an intrinsically disordered region of KLF5, enabling its recruitment to lineage-specific enhancers and driving transcriptional regulation of identity-defining genes. Notably, small-molecule inhibitors of RUVBL1/2 ATPase activity, which have anti-PDAC activity in vivo, suppress KLF5-dependent transcription. These findings define a previously unrecognized mechanism of ATP hydrolysis-dependent transcriptional coactivation and highlight a potential therapeutic strategy for modulating aberrant lineage programs in cancer.

Cellular plasticity and epigenetic reprogramming are prominent features of many human cancers[1]. In epithelial cancers (also known as carcinomas), tumor cells can transition between glandular, basal, neuroendocrine, and mesenchymal cell states, each conferring distinct fitness advantages during tumor evolution and metastasis[2]. Relevant to this study, basal lineage features, characterized by transcriptional and/ or histological resemblance to cells of stratified squamous epithelium (e.g., epidermis and esophagus)[3–6], emerge in several human adenocarcinomas, including those of the bladder, breast, and pancreas. The presence of basal identity in these tumors is associated with inferior clinical outcomes and differential responses to both conventional chemotherapy and targeted therapies[7–13]. Functional experiments have further demonstrated that the activation of basal identity programs directly promotes more aggressive disease characteristics[14–16]. In addition, acquisition of basal identity can be a mechanism of evading oncogene-targeted therapies, such as inhibitors of KRAS and EGFR in lung adenocarcinoma[17,18]. For these reasons, the biochemical mechanisms that specify basal identity are of high interest to the cancer research field.

Pancreatic ductal adenocarcinoma (PDAC) is an aggressive and heterogeneous malignancy that exemplifies the clinical significance of lineage identity. PDAC tumors exhibit epigenetic and transcriptomic plasticity, giving rise to distinct cellular states that correlate with histopathology and clinical prognosis[13,19]. Two prominent cellular identities in PDAC, termed the 'classical' and 'basal-like' states, have been well-validated and tend to exist in a mutually exclusive manner[10–13,20]. The classical state features high expression of endodermal transcription factors (TFs) (e.g., HNF4α) and lineage markers (e.g., MUC1 and MYO1A), whereas the basal-like state (also known as squamous) features high expression of the TF ΔNp63 and

[1]Cold Spring Harbor Laboratory, Cold Spring Harbor, NY, USA. [2]Cold Spring Harbor Laboratory School of Biological Sciences, Cold Spring Harbor, NY, USA. [3]Molecular and Cell Biology Graduate Program, Stony Brook University, Stony Brook, NY, USA. [4]Sidney Kimmel Comprehensive Cancer Center, Johns Hopkins University, Baltimore, MD, USA. ✉e-mail: vakoc@cshl.edu

basal lineage markers (e.g., KRT5 and KRT17)[21–23]. Across several independent patient cohorts, the expression of basal lineage markers correlates with poor overall survival, associated with increased metastatic potential and resistance to cytotoxic chemotherapy[13,24]. In accord with these clinical correlations, laboratory studies have shown that activation of basal identity (e.g., by ectopic ΔNp63 expression) drives more aggressive PDAC tumors[22,23,25].

KLF5 is a zinc finger-containing TF oncoprotein that regulates epithelial lineage identity in the aerodigestive tract and in the epidermis[26–29]. While a strong KLF5 requirement exists to complete embryogenesis and for wound healing responses in adult tissues[29–31], conditional knockout studies demonstrate that KLF5 is dispensable for homeostasis of epithelial cells in the adult lung, intestine, and pancreas[32–34]. In contrast, KLF5 is a critical dependency in multiple carcinomas, as shown in both human cancer cell lines and genetic mouse models[35–37]. In mouse models of PDAC, *Klf5* is transcriptionally upregulated during the acinar-to-ductal cell fate transition triggered by inflammation and by mutant KRAS[34,38]. In addition, KLF5 has been found to maintain classical identity in human PDAC cell lines, but its role in basal-like PDAC remains undefined[39].

RUVBL1 and RUVBL2 are evolutionarily conserved AAA+ ATPases that form obligate hetero-hexamers, which couple ATP hydrolysis to the regulation of protein-protein interactions and protein folding[40,41]. For example, they function as core scaffolding subunits for the assembly of multiprotein complexes, including the R2TP protein chaperone complex and the INO80 family of chromatin remodeling complexes (INO80, SRCAP, and TIP60/p400)[42–45]. In addition, emerging evidence suggests that RUVBL1/2 may also participate in transcriptional regulation through interactions with RNA Polymerase II and TFs such as MYC[46,47]. AAA+ ATPase activity is required for several RUVBL1/2 functions (e.g., protein chaperone activity); however, evidence also exists for ATPase-independent functions of RUVBL1/2[43,44,48].

Although RUVBL1/2 are essential proteins in all eukaryotic cells, independent laboratories have developed small molecule inhibitors of RUVBL1/2 ATPase activity, which have anti-tumor activity in pre-clinical cancer models at well-tolerated doses[48,49]. This therapeutic effect has been observed in diverse forms of cancer, including adenocarcinomas of the lung and pancreas, Ewing sarcoma, and hematopoietic malignancies[47–50]. These compounds engage an allosteric pocket at the RUVBL1/RUVBL2 interface, trapping the complex in a rigid, ATP-bound conformation[48,49,51]. This state constrains RUVBL1/2 flexibility, implicating conformational dynamics linked to ATP hydrolysis as critical for their function[51]. The observed therapeutic effects of RUVBL1/2 inhibitors in cancer-bearing mice raise the possibility that this AAA+ ATPase activity regulates oncoprotein function. However, the precise mechanisms by which RUVBL1/2 supports oncogenic pathways remain poorly defined[47,48].

Here, we identify KLF5 as a context-dependent master regulator of cell identity in PDAC, capable of activating both classical and basal transcriptional programs via a flexible cistrome. Using an integrated biochemical and genetic screening strategy, we reveal the RUVBL1/2 complex as a direct, ATPase-dependent coactivator of KLF5. Our findings suggest that RUVBL1/2 carries out this function independently of both the R2TP and INO80 family of protein complexes by binding to a disordered segment of KLF5, which enables recruitment to lineage-specific enhancers. Pharmacologic inhibition of RUVBL1/2 ATPase activity disrupts this interaction, impairs KLF5 function, and suppresses tumor cell proliferation. We also present evidence that RUVBL1/2 can function more broadly to support oncogenic TFs important in other cancer contexts. Taken together, our findings reveal a dual regulatory role for KLF5 in PDAC lineage specification and establish AAA+ ATPase-driven conformation dynamics as a previously unrecognized transcriptional coactivator mechanism with therapeutic potential.

## Results

### KLF5 is highly expressed in the classical and basal-like subtypes of human PDAC

We previously carried out genetic screens in search of novel regulators of basal identity in three independent models of basal-like PDAC (T3M-4, BxPC-3, and KLM-1), using KRT5 staining as a marker[52]. In addition to validating ΔNp63 as a master regulator of this cell state, these screens nominated KLF5 as a requirement for basal identity in each of the three PDAC models (Fig. 1A, Supplementary Data 1)[22,23]. This finding was unexpected, as previous studies demonstrate that KLF5 activates classical PDAC lineage identity, but suggest that KLF5 is absent in 'high-grade' PDAC tumors and is downregulated during epithelial-to-mesenchymal transitions (EMT)[39,53]. To investigate a possible role for KLF5 in basal-like PDAC, we re-analyzed several bulk RNA-seq datasets of human PDAC samples[10–13,20,54]. We found that *KLF5* was elevated in PDAC tumors and metastases relative to normal pancreas tissue, with similar levels in both classical and basal-like tumor subsets (Figs. 1B, C, S1A–E, Supplementary Data 2). Re-analysis of single-cell RNA-sequencing data from a genetically engineered PDAC mouse model[55] further confirmed *Klf5* upregulation during disease initiation, and demonstrated that *Klf5* expression is sustained throughout disease progression (Figs. 1D, S1F). Moreover, we re-analyzed single-nucleus RNA sequencing (snRNA-seq) data from 224,988 PDAC cells isolated from 43 resected primary human tumors to evaluate the intra-tumoral heterogeneity of *KLF5* expression (Figs. 1E, S1G)[56]. Classical and basal identities are present as distinct cell states across this set of tumors, which express *HNF4A* or *TP63*, respectively, at high levels (Figs. 1E, F, S1H). Importantly, *KLF5* was expressed at comparable levels in both subtypes of PDAC cells and was increased relative to normal pancreatic epithelial cells (Fig. 1F). In human PDAC cell lines, we found that *KLF5* was also highly expressed, particularly in models with strong classical or basal identity (Figs. 1G, S1I). Unlike primary human PDAC tumors, we found that many human PDAC cell lines weakly express both the classical and basal signatures, with a subset of such models expressing *KLF5* at low levels (e.g., PANC-1 cells) (Figs. 1G, S1I, Supplementary Data 3). Taken together, these observations indicate that KLF5 is expressed at high levels in both the classical and basal-like subtypes of human PDAC tumors and cell lines, which prompted us to investigate further the function of KLF5 in these two lineage contexts.

### Distinct KLF5 cistromes and output in classical and basal-like PDAC models

We used CRISPR-Cas9 to inactivate *KLF5* in two classical (AsPC-1 and SUIT-2) and two basal-like (T3M-4 and BxPC-3) PDAC models, followed by RNA-sequencing. While a common set of target genes was reduced in expression following *KLF5* knockout across these four lines, we observed substantial heterogeneity across these models (Fig. S2A–C, Supplementary Data 4). In accord with a prior study[39], we found that KLF5 is essential to maintain classical identity in AsPC-1 and SUIT-2 cells (Fig. S2A, C). Consistent with our KRT5 marker-based screening results (Fig. 1A), we found that *KLF5* knockout in T3M-4 and BxPC-3 cells led to a marked downregulation of basal identity genes, including *KRT5*, *KRT6A*, *KRT14*, *KRT17*, and *SPRR1B* (Fig. S2B, C). Gene Ontology analysis and Gene Set Enrichment Analysis (GSEA) further confirmed that a prominent output of KLF5 in T3M-4 and BxPC-3 cells is to maintain expression of basal identity genes in PDAC (Fig. S2B, D).

We next evaluated whether activation of classical and basal identity genes is a direct function of KLF5 in PDAC. To address this, we coupled an inducible KLF5 degron with thiol (SH)−linked alkylation for the metabolic sequencing of RNA (SLAM-seq), which is a strategy for defining the direct transcriptional effects of a TF[57,58]. Importantly, SLAM-seq measures effects on newly synthesized RNA, which can be distinguished from effects on total mRNA levels (which can have

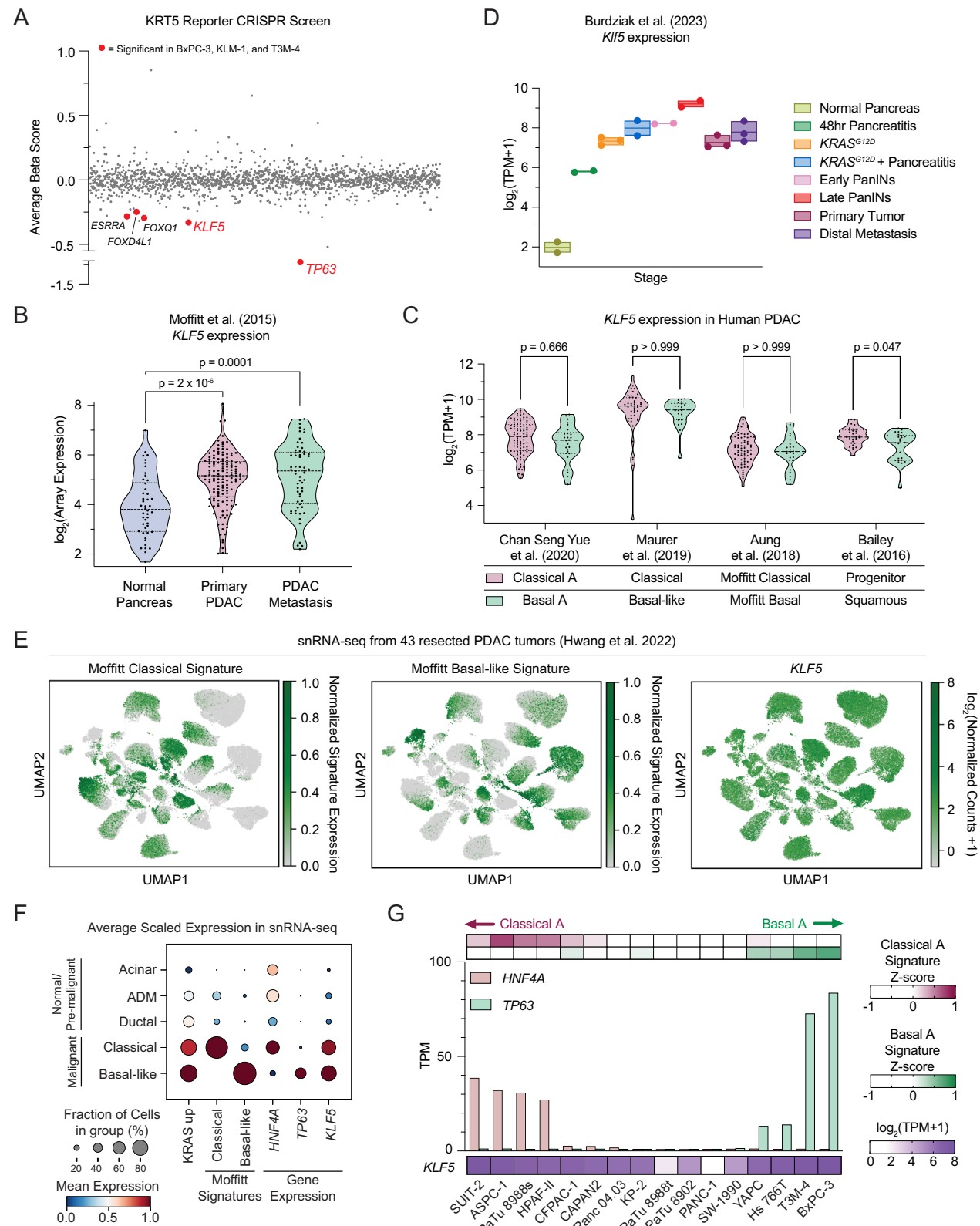

**A** KRT5 Reporter CRISPR Screen

**D** Burdziak et al. (2023) *Klf5* expression

**B** Moffitt et al. (2015) *KLF5* expression

**C** *KLF5* expression in Human PDAC

**E** snRNA-seq from 43 resected PDAC tumors (Hwang et al. 2022)

Moffitt Classical Signature | Moffitt Basal-like Signature | *KLF5*

**F** Average Scaled Expression in snRNA-seq

**G**

longer and more variable half-lives). We replaced endogenous KLF5 with an FKBP12^F36V-tagged KLF5 in both AsPC-1 and T3M-4 PDAC cells, allowing for dTAG^v-1-mediated degradation (Fig. 2A)[59]. After validating that efficient KLF5 degradation occurred within 2 h of dTAG^v-1 treatment, we performed SLAM-seq to compare KLF5 function in classical versus basal-like PDAC models (Figs. S2D–J, S3). This analysis revealed that KLF5 degradation led to a rapid and pronounced suppression of

marker genes and transcriptional signatures that define classical and basal identity in AsPC-1 and T3M-4 cells, respectively (Figs. 2B–D, S2D–J, Supplementary Data 5). Collectively, these data indicate that KLF5 performs a direct transcriptional function in PDAC that activates both classical and basal identity genes.

We next compared the genome-wide occupancy of KLF5 in classical versus basal-like PDAC models. Chromatin immunoprecipitation

**Fig. 1 | KLF5 is highly expressed in the classical and basal-like subtypes of human PDAC. A** Genome-wide KRT5 reporter CRISPRi screens in three basal-like PDAC cell lines (KLM-1, T3M-4, and BxPC-3)[52]. Beta scores and significance were calculated using MAGeCK[85] (maximum likelihood estimation). Negative beta scores indicate enrichment in the KRT5[low] population. Average beta scores of 1599 transcription factors are shown, ordered alphabetically along the x-axis. TFs with a *p*-value < 0.01 in all three cell lines labeled in red. **B, C** *KLF5* expression in resected human PDAC tissue. Statistical significance was evaluated using two-sided Mann−Whitney tests with multiple comparison corrections (**B**, 3; **C**, 4). **B** *KLF5* microarray expression from 46 normal pancreas, 145 primary tumor, and 61 PDAC metastasis samples, reanalyzed from Moffitt et al.[12]. **C** RNA-seq data reanalyzed from Chan Seng Yue et al.[13], Maurer et al.[20], Aung et al.[54], or Bailey et al.[11]. Individual samples were assigned to Classical/Progenitor or Basal/Squamous according to the classifications in each respective study. *KLF5* expression ($\log_2$(*KLF5* TPM + 1)) is plotted. **D** Single-cell RNA-sequencing of mouse pancreatic epithelial cells from Burdziak et al.[55], reanalyzed in bulk by disease stage. *Klf5* expression ($\log_2$(*Klf5*

TPM + 1)) is plotted. Each dot = one sample. Box = range. Line = mean. **E, F** Single-nucleus RNA-sequencing (snRNA-seq) from 43 resected human PDAC tumors reanalyzed from Hwang et al.[56]. **E** UMAP of malignant nuclei (dots), colored by the normalized signature expression or *KLF5* expression ($\log_2$(Normalized counts)). **F** Dot-plot of all acinar, acinar-to-ductal metaplasia (ADM), ductal, and malignant cells, colored by *HNF4A*, *TP63* or *KLF5* expression ($\log_2$(Normalized counts)), or mean normalized expression across all genes in the listed signature. Classical and Basal-like malignant populations were defined by classical_score > −0.125, basal_-score < 0.05 (classical) or classical_score < −0.125, basal_score > 0.05 (basal). **G** Gene expression in human PDAC cell lines (CCLE). Bar chart shows *HNF4A* and *TP63* expression in each cell line, ordered by *HNF4A* expression − *TP63* expression. Heatmap (top) shows median expression of Basal-A or Classical-A identity signature[13] genes (Z-score analysis of variance-stabilized transformed counts across cell lines). Scale bars indicate Z-scores. Heatmap (bottom) shows *KLF5* expression ($\log_2$(*KLF5* TPM + 1)). Source data are provided as a Source Data file.

sequencing (ChIP-seq) revealed that KLF5 binds to 2025 shared regulatory elements present in both categories of PDAC cell lines, which tend to be enriched for promoter regions (Figs. 2E, S4A). In addition, we classified 818 KLF5 peaks as being classical PDAC-specific and 1551 peaks as being specific to basal-like PDAC, with both sets more enriched for distal (intergenic and intragenic) locations, suggesting they are lineage-specific enhancers (Figs. 2E–G, S4A, Supplementary Data 6). Using H3K27 acetylation as a marker of cis-regulatory element (CRE) activity, we employed the acute degradation system and found that KLF5 was functional at all three classes of CREs (Fig. 2F–I). As examples, we highlight the classical PDAC-specific gene *HNF4A* and the basal-like PDAC-specific gene *KRT17* (Fig. 2F, G). Importantly, we observed a correlation between the KLF5 cistromes and the direct transcriptional target genes identified using acute KLF5 degradation-SLAM-seq analysis (Fig. S4B, C). Classical PDAC-specific KLF5 binding sites were enriched for recognition motifs and occupancy of HNF4α, a known master regulator of this tumor subtype (Figs. 2J, S4D–F, Supplementary Data 7)[21]. In contrast, basal-like PDAC-specific KLF5 binding sites were enriched for recognition motifs and genomic occupancy of ΔNp63, a master regulator of basal identity (Figs. 2K, S4D–F, Supplementary Data 7)[22,23,25]. Taken together, these data suggest that a flexible KLF5 cistrome supports two distinct lineage identity programs in PDAC, operating in cooperation with other lineage-defining TFs.

HNF4α and ΔNp63 are essential for the growth of classical and basal-like PDAC cell lines, respectively[23,60,61]. This prompted us to investigate whether KLF5 is also a dependency in these two forms of PDAC. Using competition-based proliferation assays, we found that KLF5 is essential for the proliferation of most PDAC cell line models, particularly those harboring robust classical or basal identity (Figs. 2L, S5A, B, Supplementary Data 3). An analysis of the Cancer Dependency Map[61] further supported this observation (Fig. S1I). 4 of the 17 PDAC cell lines were not dependent on KLF5, but these PDAC cell lines also exhibit weak expression of classical and basal transcriptional signatures, in association with low *KLF5* expression (Fig. S5A). As described above, *KLF5*-low PDAC cell lines do not appear to resemble a common cellular identity present in human PDAC tumors (Fig. 1E, F). We next orthotopically transplanted *KLF5* knockout and control AsPC-1 (classical) or T3M-4 (basal-like) cells into the pancreas of immunodeficient mice (Fig. 2M). In both tumor models, *KLF5* knockout reduced tumor growth and extended animal survival, confirming its requirement for tumor formation (Figs. 2N, O, S5C–F). Endpoint tumor analysis revealed that KLF5 function was restored in each tumor despite effective knockout at the time of transplantation (Figs. 2P, S5G–J). This suggests that *KLF5* knockout tumors select for the outgrowth of rare cells (non-edited or with in-frame changes) in the bulk population (Fig. S5H, I). Collectively, these results support that KLF5 is a genetic dependency in both classical and basal-like models of PDAC.

## An integrated proteomic-genetic screening strategy reveals RUVBL1 and RUVBL2 as KLF5 coactivators

We next sought to understand the biochemical mechanism by which KLF5 regulates lineage identity in PDAC, focusing on proteins that function as coactivators of KLF5-dependent transcription. For this purpose, we employed a screening strategy to reveal KLF5 interacting partners, using proteomics, that are also essential for expression of a KLF5-dependent transcriptional reporter in PDAC cells, using a marker-based CRISPR KO screen. For the proteomic screen, we immobilized recombinant KLF5 protein on magnetic beads, followed by a pulldown using PDAC cell nuclear lysates and mass spectrometry. This analysis identified 359 significantly enriched KLF5 interactors (Fig. S6A–C, Supplementary Data 8). For the reporter screen, we used the epigenomics datasets described above to nominate *SERPINB5* as a robust downstream KLF5 target gene, chosen because of its relevance in both classical and basal-like PDAC models (Figs. 2C, D, 3A, S2A–C, H, S6D–G). By performing antibody staining of Serpin B5 protein in fixed PDAC cells, we established conditions for flow cytometry-based analysis of this marker and confirmed the KLF5 requirement for Serpin B5 expression using this assay (Figs. 3B, S6H, I). Using the screening strategy depicted in Fig. 3C, we first carried out a knockout screen of all human TFs in AsPC-1 cells, which confirmed that KLF5 is the dominant TF required for Serpin B5 expression (Fig. S6J, K). We then expanded this screening approach to a genome-wide scale; *SERPINB5* and *KLF5* ranked as the #1 and #2 hits, respectively, confirming the overall accuracy and sensitivity of the screen (Figs. 3D, S6L). While the reporter screen recovered hundreds of genes required for Serpin B5 expression, including previously reported KLF5 cofactors such as EP300[62], only a small fraction encoded proteins that physically interact with KLF5 (Fig. 3D, Supplementary Data 1). Many general transcriptional coactivators, such as MED12, TADA2B, and EP300 did not associate with KLF5, despite being required for Serpin B5 expression (Fig. 3D, E). Instead, our integrated screening approach revealed RUVBL1 and RUVBL2 as top candidates that both bind to KLF5 and are required for expression of Serpin B5 (Figs. 3D, S7A, Supplementary Data 1, 8).

RUVBL1 and RUVBL2 are AAA+ ATPase proteins with known roles in transcriptional regulation via their presence in INO80 family chromatin remodeling complexes[40]. However, subunits of these complexes were not identified in our mass spectrometry analysis as KLF5-associated proteins, nor did they score in the Serpin B5 reporter screen (Fig. S7B, C). Similarly, core R2TP subunits (RPAP3 and PIH1D1) were not detected in either screen (Fig. S7B, C). We performed validation experiments which showed that CRISPR-mediated knockout of *RUVBL1* or *RUVBL2* led to reduced Serpin B5 expression, as measured by both western blotting and flow cytometry (Figs. 3F, S7D). RNA-seq analyses in classical and basal-like PDAC models further demonstrated that knockout of either *RUVBL1* or *RUVBL2* suppressed a broader

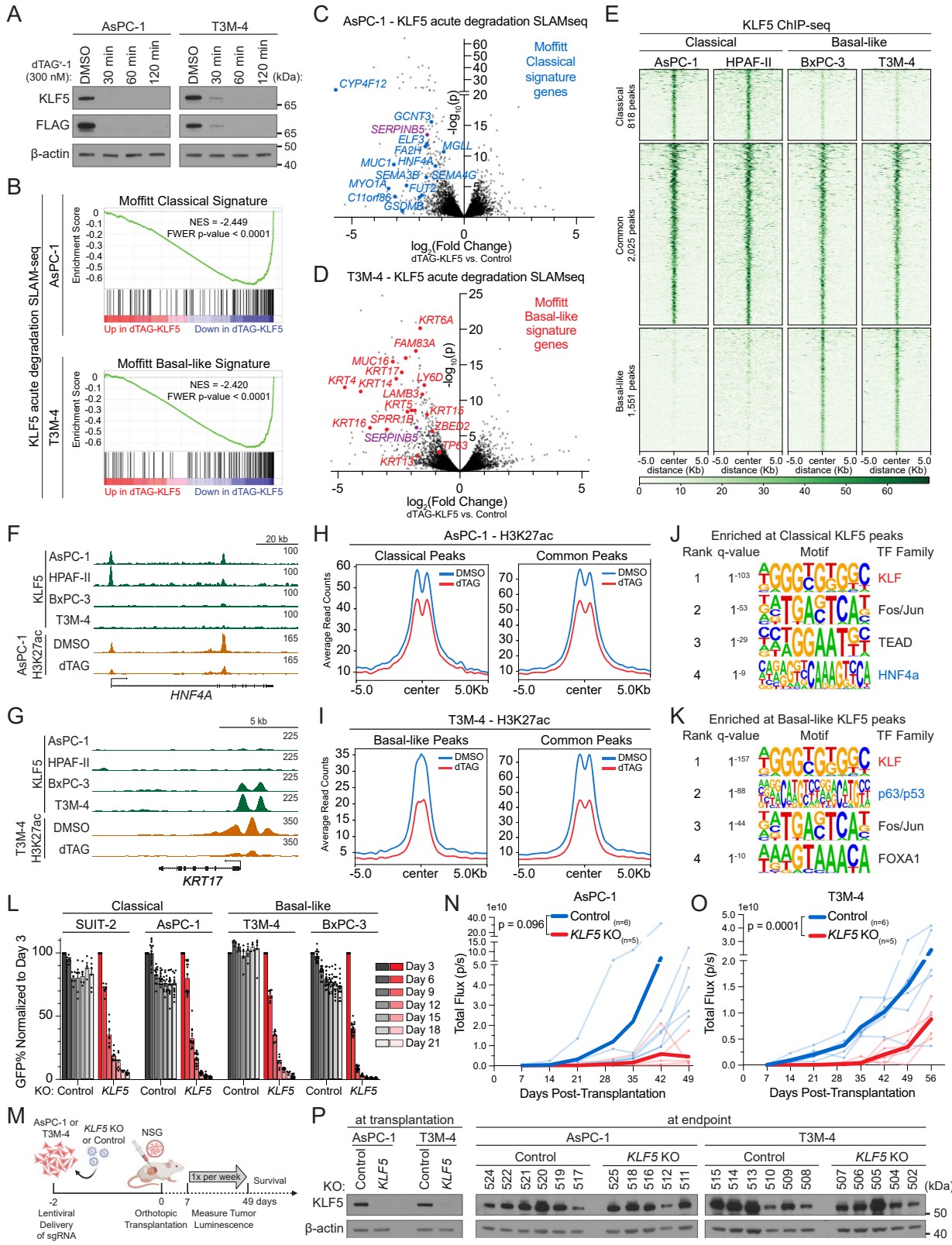

program of KLF5-dependent gene expression (Figs. 3G, H, S7E, F, Supplementary Data 9). In addition, we used immunoprecipitation-western blotting to confirm that KLF5 associates with RUVBL1/2, but not with components of INO80-family or R2TP complexes (Fig. 4A). Together, these findings prompted us to investigate RUVBL1/2 as a KLF5 coactivator, functioning independently of its established role in chromatin remodeling complexes.

## KLF5 binds to RUVBL1/2 via a functionally important disordered region

To investigate whether RUVBL1/2 binds to KLF5 directly, we expressed and purified RUVBL1, RUVBL2, and KLF5 in *E. coli* cells and evaluated for interactions using pulldown assays (Fig. S8A, B). These experiments demonstrated that KLF5 interacts with the RUVBL1/2 complex, but not with either subunit alone (Fig. 4B). KLF5 is a highly disordered protein,

**Fig. 2 | KLF5 is a lineage master regulator of classical and basal lineage identity in PDAC. A** Western blot of KLF5 in AsPC-1 and T3M-4 in which endogenous KLF5 is replaced with 3xFLAG-FKBP12$^{F36V}$-KLF5. Whole-cell lysates collected following 300 nM dTAGv-1 treatment for indicated times. Representative of 3 biological replicates. β-actin, loading control. **B–D** SLAM-seq[92] following 4-h total treatment with 300 nM dTAG$^v$-1 or dTAG$^v$-1-NEG (control), including 2-h 4sU labeling. Representative of 2 (T3M-4) or 3 (AsPC-1) biological replicates. Fold change and significance calculated by DESeq2[87]. **B** Gene set enrichment analysis[89] of differentially expressed transcripts following dTAG$^v$-1. Significance by GSEA. NES normalized enrichment score, FWER family-wise error rate. **C, D** Volcano plots of differentially expressed transcripts following dTAG$^v$-1. Select Classical and Basal-like genes (Moffitt et al.[12]) and *SERPINB5* labeled. **E–K** KLF5 ChIP-seq in two classical (AsPC-1 and HPAF-II) and two basal-like (BxPC-3 and T3M-4) lines. Classical, Basal-like, and Common sets defined by shared peaks (MACS2[90] q < 0.01, bedtools[95] intersect). *n* = 2 independent IPs per line. **E** Heatmap of 4394 KLF5 peaks. Rows = 10Kb genomic regions centered on a KLF5 peak, ordered by KLF5 signal. **F, G** Representative KLF5 and H3K27ac ChIP tracks, visualized in UCSC genome browser[100]. Matched, scaled track heights indicated (right). **H, I** H3K27ac ChIP-seq following 3.5-h dTAG$^v$-1 or DMSO (control) treatment. Metagene plots show average H3K27ac signal at all KLF5 peaks in each peak set. **J, K** HOMER[91] motif enrichment for subtype-specific KLF5 peaks. The top 4 transcription factor family motifs were selected. **L** Competition-based fitness assays in Cas9-expressing SUIT-2, AsPC-1, T3M-4, and BxPC-3 cells after CRISPR-Cas9 knockout (KO) of *KLF5* or *ROSA26* (control) using sgRNAs coupled to GFP. Bars = mean ± SD of normalized %GFP (to day 3 post-infection). Two independent sgRNAs per KO. *n* = 3 (SUIT-2, T3M-4) or 5 (AsPC-1, BxPC-3) biological replicates. Significance reported in Source data. **L, N, O** Significance assessed by Generalized Additive Mixed Model. **M** Schematic of orthotopic transplantation of luciferase+ human PDAC cell lines. Created in BioRender. Cunniff, P. (2025) https://BioRender.com/8362080. **N–P** *n* = 5–6 mice per group. **N, O** Bioluminescence of orthotopic PDAC tumors. Lighter lines: individual mice. Dark line: mean(log$_{10}$(luminescence)). **P** Western blot of KLF5 after *KLF5* or *ROSA26* (control) KO (day 5) or from freshly resected primary tumors. β-actin, loading control. Representative of two Western blot replicates. Source data are provided as a Source Data file.

with only the C-terminal triple zinc-finger DNA-binding domain (DBD) having a predicted structure (Figs. 4C, S8C)[63]. Using a deletion analysis, we identified a segment within the KLF5 intrinsically disordered region (IDR) comprising amino acids 92–182 (IDR2) as both necessary and sufficient for the RUVBL1/2 interaction (Figs. 4D, E, S8D). As a control, we found that RUVBL1/2 did not associate with its paralog KLF4, which contains a divergent IDR sequence compared to KLF5 (Fig. S8E, F)[64]. Using this assay, we also detected the known interaction between RUVBL1/2 and MYC[47], as well as interactions with other cancer-promoting TFs, including POU2F3 and MYB (Fig. S8F). To evaluate the functional relevance of the KLF5 IDR2 segment, we performed gene-complementation assays in AsPC-1 cells following CRISPR-based inactivation of endogenous *KLF5*. This experiment revealed that deletion of either IDR2 or the zinc finger DNA-binding domain rendered KLF5 incapable of rescuing its essential function in PDAC (Figs. 4F, S8G). Together, these results indicate that a functionally important IDR2 of KLF5 mediates a direct and essential interaction between KLF5 and RUVBL1/2. In addition, our findings also suggest a broader capacity of RUVBL1/2 to interact with multiple cancer-relevant TFs.

We next performed ChIP-seq analysis to profile RUVBL1/2 chromatin occupancy in classical and basal-like PDAC models. ChIP-seq analysis of RUVBL1 and RUVBL2 with independent antibodies revealed co-occupancy at approximately 95% of binding sites (Fig. S9A). A motif enrichment analysis of RUVBL1/2 peaks revealed KLF recognition motifs among the top correlates of RUVBL1/2 genomic occupancy, in addition to motifs recognized by other TFs (e.g., NFY, ETS, and MYC) (Figs. 4G, S9B, Supplementary Data 7). As observed with KLF5, we found that a significant proportion of RUVBL1/2 peaks were specific to classical or basal PDAC-like models (Fig. S9C–E). Moreover, RUVBL1/2 occupancy overlapped extensively with KLF5-bound genomic sites in both PDAC contexts. (Figs. 4H, I, S9C, F–H). RUVBL1/2 peaks tended to be broader than those of KLF5, which is in accord with our observation that RUVBL1/2 associates with multiple TFs, and suggestive of additional factors that may contribute to RUVBL1/2 recruitment at KLF5-occupied CREs. (Figs. 4H, S9H).

To determine whether KLF5 is required for RUVBL1/2 recruitment to chromatin, we profiled RUVBL1/2 genomic occupancy following acute KLF5 degradation in classical and basal-like PDAC models. Acute degradation of KLF5 resulted in specific reductions in RUVBL1 occupancy at KLF5-bound sites, in agreement with a biochemical interaction existing between these proteins on DNA (Figs. 4H, I, S9I–K). As examples, we observed KLF5-dependent RUVBL1 occupancy at the *SERPINB5* locus and at CREs at lineage identity genes, such as *HNF4A*, *MUC1*, and *FA2H* in classical PDAC and *KRT5*, *FAM83A*, and *TAF1D* in basal-like PDAC (Figs. 4I, S9L, M). In agreement with our biochemical data, ChIP-seq analysis revealed that acute KLF5 degradation did not alter the association of INO80 with the PDAC genome (Fig. S9N). Taken

together with our biochemical experiments, these findings support a model in which KLF5 recruits RUVBL1/2 to chromatin in PDAC cells, including at lineage-specific genes that define both classical and basal-like cellular identities.

## The AAA+ ATPase activity of RUVBL1/2 is required for its association with KLF5

We next evaluated whether the AAA+ ATPase activity of RUVBL1/2 regulates its association with KLF5. Using co-IP experiments, we compared wild-type RUVBL1/2 with Walker B mutations of both subunits (RUVBL1$^{E303Q}$ and RUVBL2$^{E300Q}$), which are defective in hydrolyzing ATP[51]. While the RUVBL2 mutation had no impact on KLF5 binding, the RUVBL1 mutation severely diminished its interaction with KLF5 (Fig. 5A). We further investigated this result using CB-6644, an allosteric inhibitor of RUVBL1/2 ATPase activity[49]. Consistent with our genetic results, CB-6644 reduced the RUVBL1/2 interaction with KLF5 in a dose-dependent manner (Figs. 5B, S10A). Likewise, we observed that CB-6644 also diminished the association of RUVBL1/2 with MYC, POU2F3, and SOX10 (Fig. S10B). While both RUVBL1 and RUVBL2 are pan-essential genes, we also observed a significant, albeit modest, difference in the sensitivity of PDAC cell lines to CB-6644 and to RUVBL1 knockout when comparing KLF5-dependent versus KLF5-independent models (Fig. S10C, D). Together, these results suggest that the interaction between KLF5 and RUVBL1/2 is regulated by the AAA+ ATPase activity of RUVBL1.

We next used genomic approaches to evaluate whether CB-6644 affects KLF5 function in PDAC cells. ChIP-seq analysis revealed that CB-6644 treatment led to a global release of RUVBL1/2 from the genome, including from KLF5-occupied sites (Figs. 5C, D, S10E). In contrast, KLF5 occupancy on chromatin was largely unaffected by CB-6644 treatment, suggesting that the interaction with RUVBL1/2 was not required for stable DNA binding by KLF5 in cells (Fig. S10F). We next performed a ChIP-seq analysis of H3K27 acetylation in PDAC cells following CB-6644 exposure, which revealed specific CREs with reduced levels of this active chromatin mark. Remarkably, the KLF5 recognition motif was the top sequence correlate of CB-6644-sensitive CREs (Figs. 5E, S10G, H). Consistent with this observation, KLF5-dependent CREs (defined using acute KLF5 degradation) also show reduced H3K27ac following CB-6644 treatment (Figs. 5F, S10I, J, Supplementary Data 10). Notably, the DNA motif recognized by MYC was less enriched at CB-6644-sensitive CREs. Together, these data support that a direct consequence of RUVBL1/2 inhibition is attenuation of KLF5 transcriptional activity.

To further investigate whether CB-6644 suppresses KLF5 function, we performed SLAM-seq analysis following 4 h of inhibitor treatment. Using multiple analytical approaches, we found that CB-6644 led to significant suppression of direct KLF5 target genes

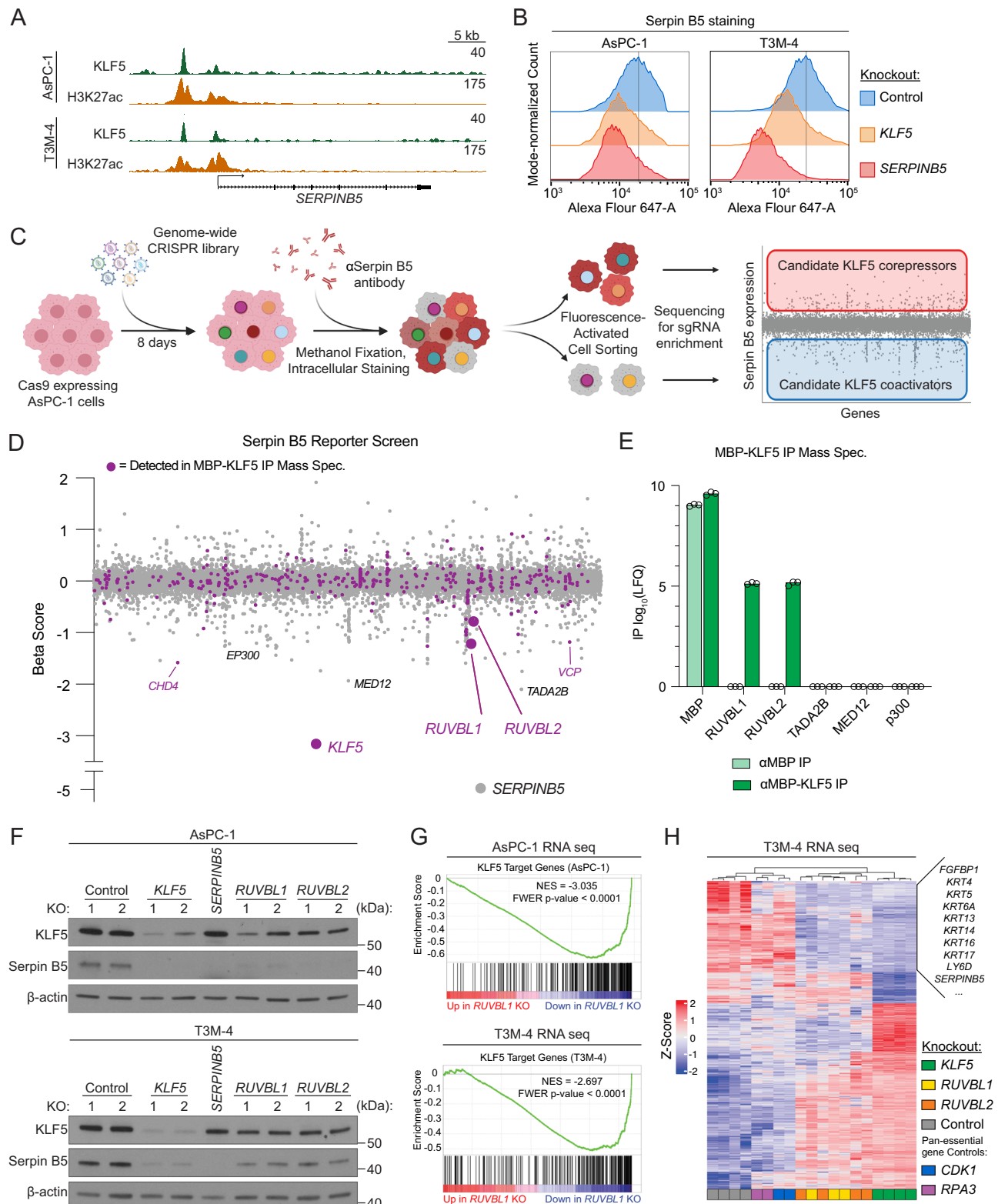

(Figs. 5G, H, S11A–D, Supplementary Data 5, 9). Like KLF5 degradation, we observed that CB-6644 triggered gene-specific transcriptional changes, which are distinct from the global transcriptional changes caused by inhibiting the general elongation kinase CDK9 with NVP-2 (Fig. S11E, F)[65]. As a control, we found that expression of either RUVBL1[A62T] or RUVBL2[ΔF109] variants, both known to confer resistance to CB-6644-mediated ATPase inhibition[49], rescued both the transcriptional changes and the PDAC cell proliferation arrest caused by

CB-6644, indicating an on-target mechanism of action of this compound (Fig. S11G, H). These genomic experiments support that RUVBL1/2 functions as a gene-specific coactivator of KLF5 in PDAC cells.

Finally, we evaluated whether RUVBL1/2 inhibition with CB-6644 disproportionately suppresses lineage identity genes in classical and basal-like PDAC models. Using both RNA-seq and SLAM-seq, we found that CB-6644 suppressed classical identity genes (e.g., *ELF3*, *FA2H*,

**Fig. 3 | Integrated proteomic and reporter screens identify RUVBL1 and RUVBL2 as KLF5 coactivators. A** ChIP-seq tracks of KLF5 and H3K27ac enrichment at the *SERPINB5* locus, visualized in UCSC genome browser[100]. Matched, scaled track heights indicated (right). **B** Flow cytometry analysis of AsPC-1 and T3M-4 cells, methanol-fixed and stained with anti-Serpin B5 antibodies 7 days following CRISPR-Cas9 knockout (KO) of *SERPINB5*, *KLF5*, or control (*ROSA26*). Representative of 4 independent experiments. Gating shown in Supplementary Fig. 6H. **C** Workflow of genome-wide Serpin B5 reporter screen. Created in BioRender. Cunniff, P. (2025) https://BioRender.com/l7na7wx. Corepressors and coactivators refer to cofactors that repress or promote KLF5 transcriptional activity, respectively. **D** Genome-wide Serpin B5 reporter screen results in AsPC-1. Genes (dots) ordered alphabetically along the x-axis. MBP-KLF5 IP Mass Spectrometry interactors from AsPC-1 nuclear lysate labeled with purple dots. Select outlier genes labeled. Beta scores and significance calculated using MAGeCK (maximum likelihood estimation). Negative beta scores indicate enrichment in the Serpin B5[low] population. Gating shown in Supplementary Fig. 6I. **E** Log$_{10}$-transformed label-free quantification (LFQ) of peptides mapping to the indicated proteins by Mass Spectrometry. MBP alone is a negative control. *n* = 3 independent IPs. Significance assessed by unpaired two-tailed Student's *t*-test (reported in Source Data). **F** Western blot of KLF5 and Serpin B5. Protein lysates were collected from AsPC-1 and T3M-4 on day 7 following KO of the indicated genes or *ROSA26* (control). Representative of 2 (T3M-4) or 3 (AsPC-1) biological replicates. β-actin, loading control. **G, H** RNA-sequencing performed on day 5 following KO of the indicated genes or *ROSA26* (control). 2 independent sgRNAs were used for *KLF5*, *RUVBL1*, *RUVBL2*, and *ROSA26*. *n* = 2 per sgRNA. Fold change and significance of differentially expressed genes (DEGs) following KO by DESeq2. **G** Gene set enrichment analysis of the DEGs following *RUVBL1* KO. Significance calculated by GSEA. NES normalized enrichment score, FWER family-wise error rate. **H** Heatmap of z-scores of variance-stabilized normalized gene counts for the top 1000 DEGs following *KLF5* knockout. Columns = samples. Rows = genes. Samples were clustered using Euclidean distance (dendrogram, top). Variance stabilized transformed counts calculated with DESeq2. Select genes associated with basal-like PDAC (Moffitt et al.[12]) labeled (right). Source data are provided as a Source Data file.

*MUC1*, and *MYO1A*) and associated transcriptional signatures in AsPC-1 cells, whereas basal identity genes (*KRT4*, *KRT5*, *KRT17*, and *S100A2*) and basal signatures were suppressed by CB-6644 treatment of T3M-4 cells (Figs. 5G, I–K, S11I, J, Supplementary Data 5, 9). We found that this effect was not limited to PDAC, as CB-6644 treatment also suppressed expression of lineage marker gene signatures in colorectal cancer, tuft cell-like small-cell lung cancer, and melanoma models (Fig. S11D, K)[66–68]. As a control, treatment with the CDK9 inhibitor NVP-2 did not induce these lineage-specific effects. (Fig. S11L). Taken together, these results suggest that RUVBL1/2 cooperates with KLF5 to activate transcriptional programs that define lineage identity in PDAC, and potentially other cancer types.

## Discussion

In this study, we identified an essential role for KLF5 as a key lineage-defining transcription factor required for the maintenance of both major molecular subtypes of human PDAC. Using an integrated screening strategy, we uncovered the AAA+ ATPase RUVBL1/2 as a critical coactivator of KLF5 in both classical and basal-like PDAC, which is recruited to the genome by KLF5 to promote lineage-specific transcriptional programs. We demonstrated that the ATPase activity of RUVBL1 is essential for binding to an intrinsically disordered region of KLF5 and that small-molecule inhibitors of RUVBL1/2 ATPase activity suppress KLF5 function through an on-target mechanism. These findings highlight an enzymatic coactivator mechanism that enables context-specific TF activity and demonstrate how TF-coactivator networks can be identified using marker-based genetic screening.

A key finding in our study is the versatility of KLF5 output at distinct stages of PDAC initiation and progression. KLF5 is known to be induced during pancreatic inflammation and stabilized by mutant KRAS to drive acinar-to-ductal metaplasia[34,38]. Additionally, prior work implicates KLF5 in sustaining classical PDAC lineage identity[39]. Our work extends these observations by showing that KLF5 expression remains elevated in advanced tumors, including in the aggressive basal-like state of PDAC. Moreover, we find that the KLF5 cistrome is significantly remodeled when comparing classical and basal-like PDAC models, which is likely to account for its divergent functions across tumor states. While earlier reports suggest that KLF5 becomes silenced during TGF-β-induced EMT and in "high-grade" PDAC cell lines[39,53], our analyses indicate that a KLF5-low state is rare in human PDAC tumors and that basal-like PDAC retains robust KLF5 expression. The clinical significance of KLF5 silencing in human PDAC remains unclear but remains a possibility as a transition state in this disease. While our study focused on models with well-defined classical (HNF4A+) or basal-like (p63+/KRT5+) lineage identities, we also note that patient tumors can display intermediate or hybrid lineage features, and the role of KLF5 in such contexts remains unclear. Collectively, our observations add to an increasing body of evidence that distinguishes basal identity from mesenchymal identity in adenocarcinoma biology, with the former retaining dependencies on epithelial oncoprotein TFs like KLF5 and ΔNp63[56,69]. Considering that several normal epithelial tissues tolerate inducible KLF5 suppression under homeostatic conditions[32–34], our findings support consideration of KLF5 as a therapeutic target in human PDAC.

Like many TFs, KLF5 lacks druggable features that might allow for direct small-molecule modulation[28,64,70]. In contrast, several TF coactivators are amenable to pharmacological targeting, owing to their use of enzymatic activities or domains that read covalent modifications[71]. One challenge in the field is in mapping specific TF-coactivator interactions that drive cancer, owing to the inherently weak affinity of their association[72]. Consistent with this notion, our proteomic screen of KLF5 interacting partners reveals numerous non-functional interactions, as evidenced by the limited overlap with hits from our genetic reporter screen. To improve elucidation of TF-coactivator interactions, our study features the integrated use of proteomics and functional genomics, which can be readily applied to other TFs to reveal functionally important interactions. A key requirement for this method is the use of a transcriptional reporter, which is predominantly activated by a single TF, exemplified here by *SERPINB5* as a specific readout of KLF5 activity. Future iterations of this pipeline could be strengthened by performing proteomic analysis of endogenous TF complexes isolated from human cells, which would enable detection of interactions that depend on post-translational modifications.

Our study presents several lines of evidence that RUVBL1/2 functions as a KLF5 coactivator independently of its presence in INO80 family chromatin remodeling complexes. For example, INO80 genomic occupancy, unlike RUVBL1/2, is unaffected by acute KLF5 degradation. Additionally, KLF5 interacts with RUVBL1/2, but not other INO80 family complex subunits, in co-immunoprecipitation assays. Structural studies have found that RUVBL1/2 adopts two major conformational states dependent on nucleotide binding, in which the external DII face of the complex is either "open" or "closed"[51]. CB-6644 locks RUVBL1/2 into the closed conformation, which resembles its structural state in the INO80 complex[51]. Since CB-6644 also disrupts the RUVBL1/2 association with KLF5, we speculate that the closed conformation of RUVBL1/2 is incompatible with KLF5 binding, a finding that could be explored in the future using structural approaches. Complementary to our results, a recent study has found that RNA Polymerase II also interacts with RUVBL1/2 independently of its association with the INO80 family of complexes[46]. This leads us to propose a model in which the "open" conformation of RUVBL1/2 (not bound to ATP or chromatin remodeling complexes) dynamically bridges TFs and RNA Polymerase II to promote gene-specific transcriptional activation.

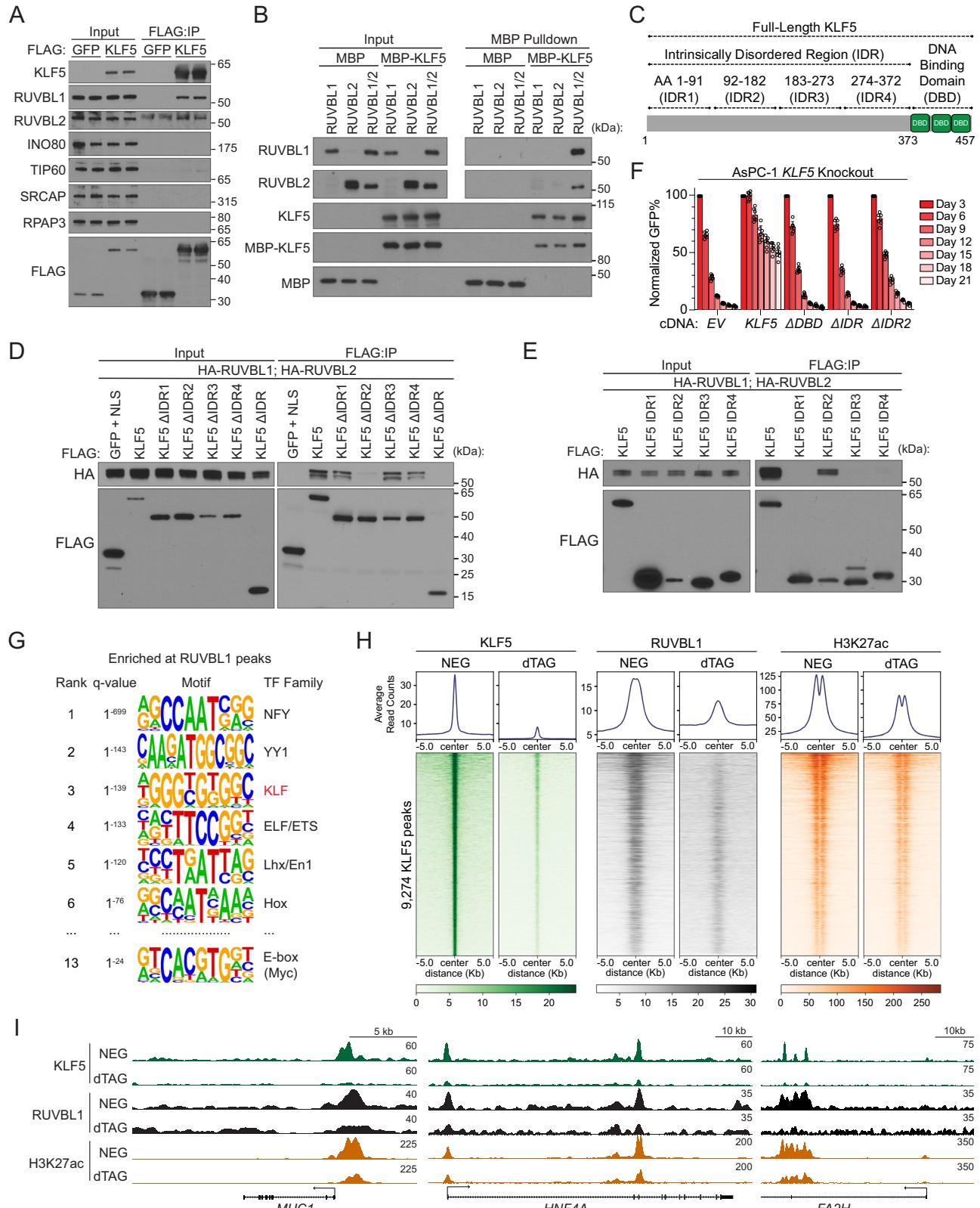

NtrC1 is a prokaryote-specific AAA+ ATPase with similarities to the functions of RUVBL1/2 in mammalian systems[73]. NtrC1 has a C-terminal sequence-specific DNA-binding domain that recognizes a motif present at enhancer elements upstream of genes transcribed by σ[54]-dependent RNA polymerase[74]. Once bound to these enhancers, NtrC1 loops to the target promoter and uses its AAA+ ATPase to promote conformational changes of σ[54]-dependent RNA polymerase that

convert it from a closed to an open conformation to drive transcription[75]. Since RUVBL1/2 lacks a sequence-specific DNA-binding domain, it has likely evolved instead a promiscuous association with disordered activation domains of transcription factors to facilitate recruitment to RNA polymerase II-dependent genes in human cells. For KLF5, this interaction is mediated primarily through the IDR2 region, although additional regions of the full-length KLF5 protein might also

**Fig. 4 | KLF5 recruits RUVBL1/2 to cis regulatory elements in PDAC cells. A** Anti-FLAG co-immunoprecipitation-Western blot. 0.5% input loaded. Two independent IP replicates per condition. FLAG-GFP, negative control. FLAG immunoblot validates equal expression of KLF5 and GFP. Representative of three biological replicates. **B** Western blot following MBP pulldown with purified, recombinant MBP-tagged KLF5 and RUVBL1/2. 1% input loaded. KLF5 and MBP, loading controls. Representative of two independent protein purifications. **A, B** Samples derive from the same experiment but different gels: **A** RUVBL1, RPAP3, SRCAP, FLAG; another for RUVBL2; another for TIP60, INO80, KLF5 or **B** RUVBL1, MBP; another for RUVBL2, KLF5 were processed in parallel. **C** Diagram of KLF5 domain architecture. **D, E** Western blot following FLAG-KLF5 and HA-RUVBL1/2 co-immunoprecipitation. Amino acid compositions of the KLF5 deletion mutants detailed in (**C**). HA immunoblot detects RUVBL1/2 pulldown. 0.5% input loaded. FLAG, loading control Representative of 2 (**E**) or 3 (**D**) biological replicates. **D** FLAG-GFP, non-interacting control. **F** Gene complementation competition-based proliferation assay of stably expressed KLF5 truncation mutants, measuring their ability to rescue endogenous

*KLF5* knockout. Data shown as mean ± SD of normalized %GFP (to day 3 post-infection). Two independent *KLF5*-targeting sgRNAs per replicate. $n = 3$ (*EV*) or 4 (all other cDNAs) biological replicates. Significance assessed by Generalized Additive Mixed Model (reported in Source data). **G** HOMER motif enrichment analysis of 6459 RUVBL1 peaks in AsPC-1 (MACS2 $q < 0.01$). The top 6 transcription factor family motifs were selected. **H, I** ChIP-seq performed in AsPC-1 cells in which endogenous KLF5 is replaced with a FKBP12$^{F36V}$-KLF5, performed 3.5 h after 300 nM dTAG$^{v}$-1 (dTAG) or dTAG$^{v}$-1-NEG (NEG, control) treatment. Spike-in normalized using mouse chromatin (FC-1199). **H** Heatmaps for KLF5, RUVBL1, and H3K27ac across all 9274 KLF5 peaks. Rows = 10 kb genomic regions centered on KLF5 peak summits, ordered by KLF5 signal in the NEG condition, and this ordering is applied to all heatmaps. Metagene plots show the average signal for each factor across all peaks, plotted above each heatmap. Representative of three biological replicates. **I** ChIP-seq tracks showing KLF5, RUVBL1, and H3K27ac enrichment at representative classical PDAC genes, visualized in UCSC genome browser[100]. Matched, scaled track heights are listed (right). Source data are provided as a Source Data file.

be involved in stabilizing the interaction with RUVBL1/2. It has been found previously that recombinant RUVBL1/2 can activate human RNA polymerase II under reconstituted in vitro conditions, but it remains unclear whether it drives conformational changes in RNA polymerase II to mediate these effects, in analogy to the NtrC1 paradigm[46]. These observations position AAA+ ATPases as an evolutionarily conserved, yet under-appreciated, class of transcriptional coactivators in human biology and disease. Structural approaches could be applied to RUVBL1/2 to further elucidate its evolutionary relatedness to NtrC1.

Several studies have demonstrated that pharmacological targeting of RUVBL1/2 can extend the survival of tumor-bearing mice, although the mechanisms that underlie a tumor-specific dependence on RUVBL1/2 ATPase activity are unclear[49–51,76]. Our study, combined with the work of others[47,48], suggests that acute transcriptional suppression is a direct effect of targeting RUVBL1/2 with small-molecule inhibitors. While KLF5 and MYC likely contribute to this phenotype in PDAC, our data suggest that other TFs are also suppressed by CB-6644. For example, both POU2F3 and SOX10 also bind RUVBL1/2 in an ATPase-dependent manner, and RUVBL1/2 inhibition broadly suppresses lineage-specific gene expression across several cancer types. While a strong possibility exists that RUVBL1/2 supports TF function in normal development, we speculate that tumor types with a strong TF dependency might be hypersensitive to RUVBL1/2 inhibition. Furthermore, this mechanism raises the possibility that inhibition of RUVBL1/2 could be a strategy to reprogram tumor cell identity into an alternative cell state that is more amenable to treatment with other therapies. For example, a RUVBL1/2 inhibitor might be combined with an oncogene-targeted therapy to prevent the emergence of a drug-tolerant lineage state[17,18,77,78]. However, it is clear from our study and others that the existing allosteric RUVBL1/2 inhibitors exert pleiotropic effects on several cellular processes, including DNA replication and chaperone activities, in addition to effects on transcription[47,48,50]. A clear opportunity for future investigation will be to identify small molecules that selectively disrupt the KLF5–RUVBL1/2 interaction, while preserving the other essential functions of the ATPase. Structural elucidation of TF-RUVBL1/2 complexes will be vital to such an effort, which may reveal unique conformations that enable the design of allosteric inhibitors with increased specificity. Nevertheless, it is remarkable that normal mouse tissues are minimally harmed by RUVBL1/2 inhibitors[47,49,50], which suggests that the essentiality of this AAA+ ATPase becomes enhanced during oncogenic transformation. These data are consistent with broader evidence that human cancer exhibits an elevated dependency on the general transcriptional apparatus[79]. Thus, a possibility exists that RUVBL1/2 is a therapeutic cancer target because it acts as a rate-limiting enzymatic coactivator required for the aberrant activity of TF oncoproteins such as KLF5.

## Methods

### Institutional approval
This study complies with all relevant ethical regulations, and all protocols were approved by the Cold Spring Harbor Institutional Biosafety Committee (IBC). Experimental protocols involving mice were approved by the Institutional Animal Care and Use Committee at Cold Spring Harbor Laboratory.

### Cancer cell lines and tissue culture
The following cell lines used in this study were obtained from ATCC: HEK293T (Cat# CRL-3216; RRID:CVCL_0063), A-375 (female, Cat# CRL-1619; RRID:CVCL_0132), AsPC-1 (female, Cat# CRL-1682; RRID:CVCL_0152), BxPC-3 (female, Cat# CRL-1687; RRID:CVCL_0186), Capan-2 (male, Cat# HTB-80; RRID:CVCL_0026), CFPAC-1 (male, Cat# CRL-1918; RRID:CVCL_1119), H1048 (female, Cat# CRL-5853; RRID:CVCL_1453), Hs 766-T (male, Cat# HTB-134; RRID:CVCL_0334), HPAF-II (male, Cat# CRL-1997; RRID:CVCL_0313), MIA PaCa-2 (male, Cat# CRL-1420; RRID:CVCL_0428), Panc 04.03 (male, Cat# CRL-2555; RRID:CVCL_1636), PANC-1 (male, Cat# CRL-1469; RRID:CVCL_0480), SW-1990 (male, Cat# CRL-2172; RRID:CVCL_1723), T84 (male, Cat# CCL-248; RRID:CVCL_0555). The following cell lines used in this study were obtained from DSMZ: PaTu-8902 (female, Cat# ACC 179; RRID:CVCL_1845), PaTu-8988s (female, Cat# ACC 204; RRID:CVCL_1846), PaTu-8988t (female, Cat# ACC 162; RRID:CVCL_1847), YAPC (male, Cat# ACC 382; RRID:CVCL_1794). The following cell lines used in this study were obtained from JCRB: KLM-1 (male, Cat#: RCB2138; RRID:CVCL_5146), KP-2 (female, Cat#: JCRB0181; RRID:CVCL_3004), SUIT-2 (male, Cat#: JCRB1094; RRID: CVCL_3172).

A-375, AsPC-1, BxPC-3, Capan-2, Hs 766-T, KLM-1, KP-2, Panc 04.03, SUIT-2, SW 1990, T3M-4, T84, and YAPC cells were cultured in RPMI-1640 supplemented with 10% fetal bovine serum (FBS). HEK293T, CFPAC-1, HPAF-II, MIA PaCa-2, PaTu-8902, PaTu-8988s, PaTu-8988t, and PANC-1 cells were cultured in DMEM supplemented with 10% FBS. H1048 cells were cultured in HITES medium, which is composed of DMEM:F12 supplemented with 0.005 mg/mL insulin, 0.01 mg/mL transferrin, 30 nM sodium selenite, 10 nM hydrocortisone, 10 nM β-estradiol, 4.5 mM l-glutamine and 5% FBS. The FC-1199 pancreatic cancer cell line was generated in the Tuveson lab using tumor tissues from the Kras$^{G12D}$Trp53$^{R172H}$Pdx1-Cre (KPC) mice of a pure C57BL/6 genetic background. FC-1199 was cultured in DMEM supplemented with 5% FBS. Penicillin–streptomycin was added to all media. Cell lines were purchased from commercial vendors and their identity validated by STR analysis. Cell lines were regularly tested for *Mycoplasma* contamination. All antibiotic concentrations used to select gene cassettes were empirically titrated in each cell line to achieve maximum selection with minimum toxicity.

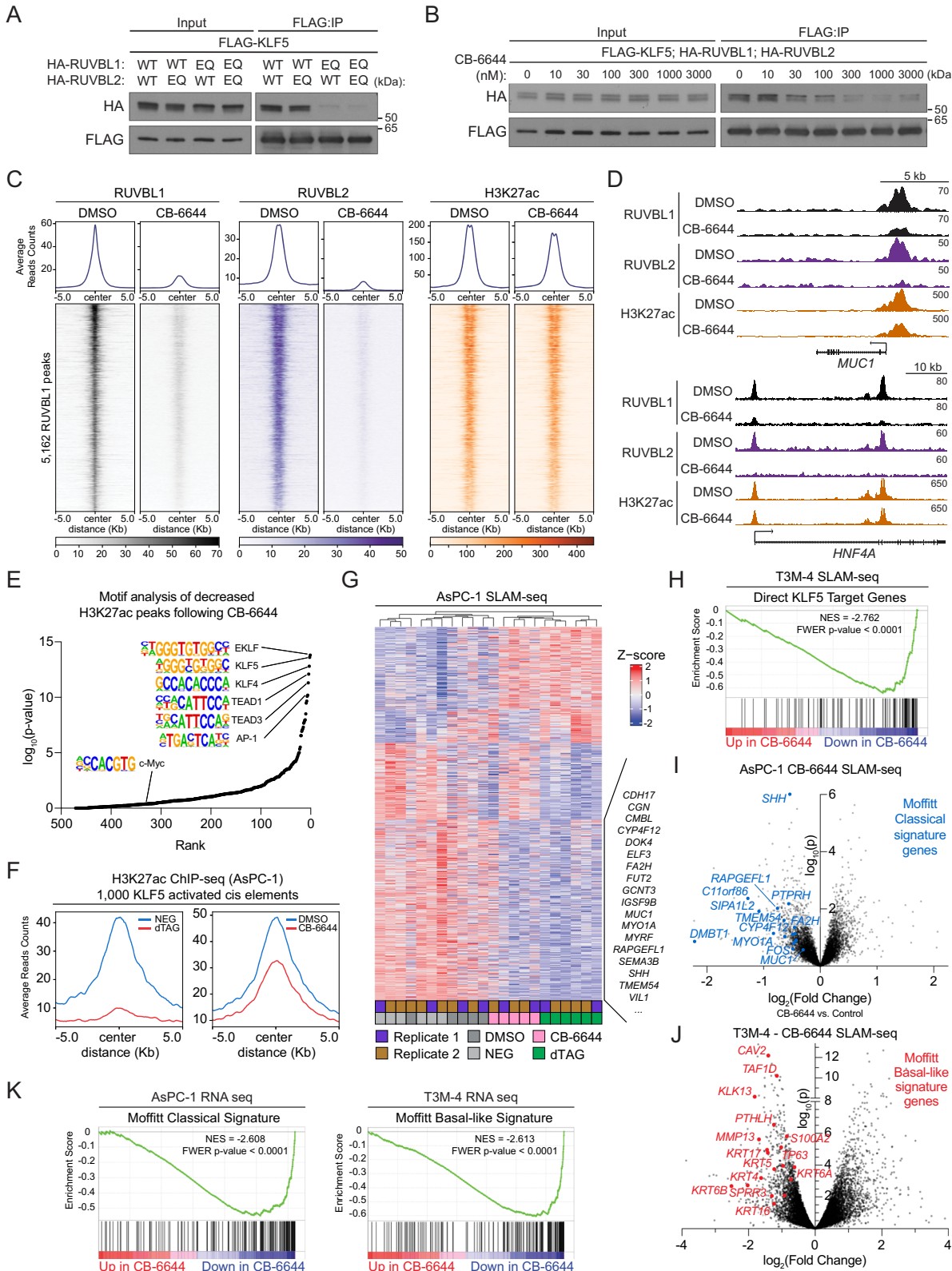

## Lentiviral production and infection

Lentivirus was produced in HEK293T cells transfected with target plasmids and packaging plasmids (VSV-G and psPAX2) using polyethyleneimine. Transfection media was replaced with fresh DMEM supplemented with 10% FBS 6 h after transfection, and lentivirus-containing supernatant was collected 24, 48, and 72 h following transfection. All three collections were pooled and filtered using a 0.45 μM PES filter. For lentiviral infections, cell suspensions were exposed to lentiviral-containing supernatant supplemented with polybrene to a final concentration of 4 μg/mL and spun at 600 RCF for 30 min. Lentiviral media was changed for fresh media after 24 h.

**Fig. 5 | Chemical inhibition of RUVBL1/2 ATPase suppresses KLF5 function and lineage identity genes in classical and basal-like PDAC. A, B** Western blot of FLAG-KLF5 and HA-RUVBL1/2 co-immunoprecipitation. HA immunoblot detects RUVBL1/2 IP. 0.5% input loaded. FLAG, loading control. Representative of three biological replicates. **A** EQ = RUVBL1/2 Walker B mutants (RUVBL1$^{E303Q}$, RUVBL2$^{E300Q}$). WT wild type. **B–K** CB-6644[49] = allosteric RUVBL1/2 AAA+ ATPase inhibitor. **C–K** Comparison of cells treated with CB-6644 (750 nM) vs. DMSO (control). **C, D** RUVBL1, RUVBL2, and H3K27ac ChIP-seq in AsPC-1 following 12-h CB-6644. **C** Heatmaps of 5162 RUVBL1 peaks (q < 0.01, MACS2; 10 kb genomic regions centered on peaks). Rows ordered by RUVBL1 signal (DMSO), and this ordering is applied to all heatmaps. Metagene plots (above) show average signal per factor. Spike-in normalized using mouse chromatin (FC-1199). Representative of two biological replicates. **D** ChIP-seq tracks of RUVBL1, RUVBL2, and H3K27ac at representative classical PDAC loci, visualized in UCSC genome browser. Matched, scaled track heights indicated (right). **E** DiffBind analysis of global H3K27ac changes in AsPC-1 after CB-6644 vs. DMSO. DESeq2 assessed enrichment. HOMER motif analysis on loci with >0.25-fold H3K27ac decrease. HOMER reported significance. Points = motifs, ranked by *p*-value. Selected motifs labeled. **F, G** dTAG = 300 nM

dTAG$^V$-1 (degrades FKBP12$^{F36V}$-KLF5), NEG = 300 nM dTAG$^V$-1-NEG (control) **(F)** H3K27ac ChIP-seq in AsPC-1 following 3.5-h dTAG, NEG or 12-h CB-6644, DMSO. Metagene plots show average H3K27ac signal at 1000 H3K27ac loci most decreased following dTAG. **G–K** SLAM-seq in AsPC-1 and T3M-4 following 4-h dTAG, NEG, CB-6644, or DMSO treatment, including 2-h 4sU labeling. *n* = 2 biological replicates per condition. Representative of 2 (T3M-4) or 3 (AsPC-1) independent experiments. **G** Heatmaps of z-scored normalized gene counts (DESeq2) for 922 differentially expressed transcripts following dTAG in AsPC-1 (FDR < 0.05). Columns = samples. Rows = genes. Z-score normalization and scaling were first performed independently (dTAG vs. NEG; CB-6644 vs. DMSO per replicate) prior to combining for clustering and heatmap generation. Samples clustered using Euclidean distance (dendrogram, top). Classical PDAC-associated genes labeled (right). **H–K** Fold change and significance of differentially expressed transcripts following CB-6644 by DESeq2. **H, K** Gene set enrichment analysis (GSEA). NES normalized enrichment score, FWER family-wise error rate. **I, J** Selected Classical and Basal-like genes (Moffitt et al.[12]) labeled. **K** RNA-seq following 24-h CB-6644. Source data are provided as a Source Data file.

## RNA extraction, RT-qPCR, and RNA-sequencing

Total RNA was extracted using TRIzol reagent following the manufacturer's instructions. For RT-qPCR, 100 ng of total RNA was reverse transcribed to cDNA using qScript cDNA SuperMix, followed by RT-qPCR analysis with SYBR green PCR master mix on an Applied Biosystems 7900HY Fast Real-Time PCR system. Primer sequences are included in Supplementary Table 1. Relative expression was calculated by normalization to *ACTB* controls followed by linearization. For RNA-seq experiments following CRISPR-based targeting of *KLF5*, *SERPINB5*, *RUVBL1*, *RUVBL2*, or controls, cells stably expressing Cas9 were infected with control or target sgRNAs in an LRG2.1_Puro vector[80] (Addgene # 125594) to >95% GFP positivity. RNA was collected at relevant timepoints as assessed by Western blot kinetics of protein depletion and loss of cell viability.

RNA-sequencing libraries were constructed using the TruSeq Sample Prep Kit V2 (Illumina) following the manufacturer's instructions. Briefly, 2 μg of extracted, purified RNA was poly-A selected and fragmented with fragmentation enzyme mix. cDNA was synthesized with SuperScript™ II reverse transcriptase, followed by end repair, A-tailing, single-end indexed adaptor ligation, and PCR amplification. RNA-sequencing libraries were single-end sequenced for 76 bp using an Illumina NextSeq platform (Cold Spring Harbor Genome Center, Woodbury, NY, 11797).

## ChIP and ChIP-seq library construction

For each ChIP, cells were trypsinized, counted, and resuspended at 5–10 × 10$^6$ cells/mL in room temperature PBS. Cell suspensions were crosslinked in 1% formaldehyde at room temperature for 15 min, followed by the addition of glycine to quench the reaction at a final concentration of 0.125 M. For experiments involving KLF5 degradation or RUVBL1/2 inhibition, dTAG$^V$-1 or CB-6644 was added to resuspension media and PBS prior to crosslinking. After two ice-cold PBS washes, cells were resuspended in cell lysis buffer (10 mM Tris pH 8.0, 10 mM NaCl, 0.2% NP-40) at 10 × 10$^6$ cells/mL and incubated on ice for 15 min. After spinning down, supernatant was removed and nuclei were resuspended in nuclear lysis buffer (50 mM Tris pH 8.0, 10 mM EDTA pH 8.0, 1% SDS) at 10 × 10$^6$ cells/mL and sonicated in 15 mL tubes using a Bioruptor® Pico water bath sonicator (30 s on/off cycles, 1 mL per tube). The number of cycles was empirically determined for each cell line to achieve an average chromatin size distribution of 200–500 base pairs, which showed that each cell line could be efficiently sonicated with 10 cycles. Each 1 mL of sonicated chromatin from 10 million cells was diluted with 7 mL of IP-Dilution buffer (20 mM Tris pH 8.0, 2 mM EDTA pH 8.0, 150 mM NaCl, 1% Triton X-100, 0.01% SDS), and 200 μL of the sample was saved for input. Chromatin from 10–20 million (H3K27ac), 40–80 million (KLF5, HNF4α, p63) or ≥120 million

(RUVBL1, RUVBL2, INO80) cells was incubated with 4 μg (H3K27ac), 6–8 μg (KLF5, HNF4α, p63), or 10 μg (RUVBL1, RUVBL2, INO80) of the appropriate antibody and 25–100 μL of magnetic protein-A (rabbit) or protein-G (mouse) beads at 4 °C overnight. Beads were then pooled for each respective IP and washed once with 1 mL IP-wash buffer 1 (20 mM Tris pH 8.0, 2 mM EDTA pH 8.0, 50 mM NaCl, 1% Triton X-100, 0.1% SDS), twice with 1 mL High-salt buffer (20 mM Tris pH 8.0, 2 mM EDTA pH 8.0, 500 mM NaCl, 1% Triton X-100, 0.01% SDS), once with IP-wash buffer 2 (10 mM Tris pH 8.0, 1 mM EDTA pH 8.0, 250 mM LiCl, 1% NP-40, 1% sodium deoxycholate), and twice with 1 mL TE buffer (10 mM Tris-Cl, 1 mM EDTA, pH 8.0). Chromatin was eluted from beads and reverse-crosslinked in 200 μL nuclear lysis buffer supplemented with 12 μL NaCl and 1 μg/mL RNase A by shaking at 800 rpm for ≥4 h at 65 °C. Supernatant was isolated from magnetic beads, and protein digestion was performed by adding 4 μg/mL of Proteinase K and incubating the mixture for 2 h at 56 °C. NaOAc pH 5.2 was added to a final concentration of 75 mM, and the DNA to be used for library prep was purified using the QIAGEN PCR purification kit and following the manufacturer's instructions. For experiments involving KLF5 degradation or RUVBL1/2 inhibition, ChIP-seq libraries were normalized by spike-in of mouse chromatin from FC-1199 cells at 10% of total chromatin prior to IP.

Each ChIP-seq library was constructed using the Illumina TruSeq ChIP Sample Prep kit, following the manufacturer's instructions. Briefly, ChIP DNA was end-repaired, A-tailed, and ligated to Illumina-compatible single-index adaptors. 12–14 PCR cycles were used for final library amplification. After amplification, the library was purified 2x with 1x Ampure XP beads and analyzed on a Bioanalyzer using a high-sensitivity DNA chip (Agilent). Library DNA concentrations were quantified using an Invitrogen Qubit 4 Fluorometer using the 1X High-sensitivity dsDNA assay. ChIP-seq libraries were single-end sequenced for 76 or 100 bp at a sequencing depth of ≥50 million raw reads (H3K27ac or input) or ≥30 million raw reads (KLF5, HNF4α, p63, RUVBL1, RUVBL2, INO80) per sample using an Illumina NextSeq platform (Cold Spring Harbor Genome Center, Woodbury, NY, 11797).

## Generation of acute degradation PDAC cell lines

Full-length KLF5 cDNA constructs containing silent mutations conferring resistance to single guide RNA (sgRNA)-targeting were cloned into a lentiviral FLAG-FKBP12$^{F36V}$-fusion expression vector containing a neomycin resistance cassette, which was used to produce lentiviral supernatant. cDNA constructs were stably expressed in AsPC-1, T3M-4, or BxPC-3 cells stably expressing Cas9 from a LentiV-Cas9-blasticidin vector and infected cells were selected with neomycin (G418) for 7–10 days. cDNA-expressing cells were then infected with sgRNAs or dual-guide RNAs (dgRNA) targeting endogenous *KLF5* in an

LRG2.1_Puro vector (Addgene # 125594). Cells were selected with puromycin for 7 days. KLF5 and FLAG Western blots were used to validate expression of FLAG-FKBP12[F36V]-KLF5, knockout of endogenous KLF5, and acute degradation of FLAG-FKBP12[F36V]-KLF5 in response to dTAG[v]-1 treatment. All experiments utilizing acute degradation of PDAC cell lines were performed at empirically determined timepoints based on the kinetics of dTAG[v]-1-induced KLF5 degradation.

## SH-linked alkylation for the metabolic sequencing of RNA (SLAM-seq)

For all experiments involving AsPC-1 and T3M-4 cells, 4sU was given at 500 μM for 2 h. Concentrations and timepoints for 4-thiouridine (4sU) were determined empirically using the SLAM-seq Explorer Kit Cell Viability Titration and S4U Incorporation Modules (Lexogen) following the manufacturer's instructions. Briefly, plated AsPC-1 or T3M-4 cells were treated with increasing concentrations of 4sU for 8 h, then subjected to cell viability analysis by CellTiter-Glo (Promega). The experimental working concentration of 4sU was determined as the concentration that causes 10% inhibition ($IC_{10}$). Separately, AsPC-1 and T3M-4 were treated for 0, 1, 2, 4, or 8 h with 500 μM 4sU and RNA collected in TRIzol and extracted as described. Extracted RNA for each sample was enzymatically digested to single nucleosides and subjected to analysis by High-performance liquid chromatography (HPLC) on a Supelco Discovery C18 reverse phase column with a size of 250 × 4.6 mm as previously described[81]. Mobile phases A (0.1 M TEAA, 3% Acetonitrile) and B (90% Acetonitrile) at 0.5 mL/min with an isocratic gradient of 100% A (15 min), 0→10% B (20 min), 10→100% B (30 min), 100% B (5 min). Concentrations of uridine and 4sU were determined from a standard curve using spike-in nucleosides at known concentrations, and the timepoint was chosen as the earliest timepoint at which 4sU achieves 0.5–1% incorporation. All cell culture, RNA extraction, and preparation of nucleosides involving 4sU were performed in the dark.

SLAM-seq experiments were performed using the SLAM-seq Explorer Kit Anabolic Kinetic Module (Lexogen) following the manufacturer's instructions. RNA from AsPC-1 or T3M-4 cells was collected in TRIzol following experimental treatment or control and 2-h treatment with 500 μM 4sU. After total RNA was extracted as described, the 4-thiol groups present on 4sU labeled transcripts were alkylated with iodoacetamide (IAA), and RNA was re-purified. Prior to IAA alkylation, all steps were performed in the dark. Libraries were then prepared from RNA using the QuantSeq 3' mRNA-Seq V2 Library prep kit (Lexogen) following the manufacturer's instructions. Briefly, 500 ng total RNA per sample is enzymatically reverse transcribed using an Illumina-compatible oligo(dT) primer. Any remaining RNA was removed, and second strand synthesis was initiated by a random Illumina compatible primer, followed by purification using magnetic beads. 15 PCR cycles were used for final library amplification. After amplification, the library was purified 1x with magnetic beads and analyzed on a Bioanalyzer using a high-sensitivity DNA chip (Agilent). Library DNA concentrations were quantified using an Invitrogen Qubit 4 Fluorometer using the 1X High-sensitivity dsDNA assay. SLAM-seq libraries were single-end sequenced for 100 bp at a sequencing depth of ≥50 million raw reads per sample using an Illumina NextSeq platform (Cold Spring Harbor Genome Center, Woodbury, NY, 11797). Spike-in controls were not included, and SLAM-seq reads were therefore not normalized to external reference standards.

## Intracellular FACS-based CRISPR screens

After empirical determination of the suitable lentiviral titer, ~5 × 10[7] (TF screen) or 7.5 × 10[8] (genome-wide screen) Cas9-expressing cells were infected with human DNA Binding Domain-Focused[82,83] (Addgene #123334) or genome-wide Brunello[84] (Addgene #73178) sgRNA library-encoding suspension for a 20–30% infection percentage. Media was changed at 48 h, and antibiotic selection was added for 72 h. 8 days

post-infection, cells were trypsinized, resuspended in serum-containing media, counted, washed in ice-cold PBS, and fixed in −20 °C methanol at ≤10 × 10[6] cells/mL under gentle vortexing. Cells were stored in methanol at −20 °C for at least 2 days and up to 1 month. One day before sorting, cells were pelleted, washed 1x in FACS buffer (1% (w/v) ultrapure BSA, 0.5% (w/v) sodium azide, and 1 mM EDTA in magnesium and calcium-free PBS), and incubated overnight in 1:200 primary antibody (Serpin B5) in FACS buffer at 10 × 10[6] cells/mL rotating at 4 °C. The next day, cells were pelleted, washed 2x with FACS buffer, and incubated for 2 h in 1:500 secondary antibody (Alexa-Fluor647-conjugated anti-mouse) in FACS buffer at 10 × 10[6] cells/mL, rotating at 4 °C protected from light. After washing 2x in FACS buffer, cells were resuspended in FACS buffer at 10 × 10[6] cells/mL and sorted. Stained cells were sorted using a BD FACS Aria II cell sorter. The total number of cells sorted per screen was a minimum of 5000x the size of the sgRNA library. Cells were sorted into three different populations, with approximately 20% of the cells sorted into the Serpin B5[low] bin, 70% of the cells sorted into the Serpin B5[bulk] bin, and 10% of the cells being sorted into the Serpin B5[high] bin. Cell pellets were then processed for DNA extraction and library preparation as described below. Custom sequencing primers were added for each respective cell population. All sequencing data from FACS-based screens were analyzed with MAGeCK[85] v0.5.9.3 using the MLE option.

## DNA extraction and sgRNA sequencing for CRISPR screens

After pooling and pelleting of sorted cells, cells were resuspended in DNA extraction buffer (10 mM Tris-HCl, pH 8.0, 150 mM NaCl, and 10 mM EDTA) at a density of ≤5 × 10[7] cells/mL. SDS and Proteinase K were added to final concentrations of 0.1% and 0.2 mg/mL, respectively. The mixture was incubated for 48 h at 56 °C, after which DNA was purified by phenol extraction. Equilibrated phenol was added 1:1 to the lysis mixture, mixed well, and centrifuged for 10 min at 20,000 RCF. Supernatant was carefully removed, and another phenol purification round was performed. DNA was then precipitated by adding 3 volumes of isopropanol and NaOAc pH 5.2 to a final concentration of 75 mM and incubating overnight at −20 °C. DNA was pelleted at 20,000 RCF for 1 h, washed in 70% ethanol, and air-dried until translucent. After resuspension in sterile, ultrapure water, DNA was assessed for quality by nanodrop before proceeding to library prep. sgRNAs were directly amplified from genomic DNA by one-step PCR using NEBNext® Ultra™ II Q5® Master Mix (NEB). Each PCR reaction was performed with 10 μg of genomic DNA in 100 μL final volume. Titrations of amplification cycles were performed for LRG or LentiCRISPRv2 (95 °C, 1 min; n cycles [95 °C, 30 s; 53 °C, 30 s; 72 °C, 30 s]; 72 °C, 10 min) sgRNA cassettes, which showed they could be efficiently amplified with 30 and 28 cycles, respectively. All PCR reactions for each sample were pooled, and 400 μL was taken for double-sided Ampure bead cleanup (0.65x + 1x) bead volume to preserve PCR amplicons (~192 and ~274 bp, respectively). Amplicons were sequenced using an Illumina NextSeq with 50% spike-in or pooled with high-diversity libraries (Cold Spring Harbor Genome Center, Woodbury, NY, 11797).

## General computational and statistical analyses

All sequencing data were analyzed using the CSHL High-Performance Computing System (HPC). Packages used to analyze next-generation sequencing data were installed in independent Anaconda environments to minimize dependency conflicts. Downstream analyses were performed using Python 3 in JupyterLab notebooks or in R Studio. Student's t-tests, Mann–Whitney U tests, linear regression calculations, Log-rank tests, and Fisher's Exact statistical tests were done using R or PRISM v10. GAMM tests were performed using the mgcv package in R.

## RNA-seq data analysis

Single-end 76 bp raw sequencing reads were pseudo-aligned to the hg38 genome using Kallisto[86] with bootstrap 100. Low abundance

transcripts were removed, and variance stabilized normalized transcripts and differential expression were calculated using DESeq2[87]. Transcripts per million (TPM) was calculated from aligned, mapped reads normalized for gene length using tximport[88].

Gene set enrichment analysis (GSEA) was conducted using GSEA_4.3.3[89]. To generate Moffitt Classical and Basal-like gene signatures, we compiled the most specific genes to each molecular subtype based on the expression data published in Moffitt et al.[12], to a maximum of 250 genes. To generate KLF5 direct target gene signatures for GSEA in AsPC-1 or T3M-4, we ranked all significantly downregulated T→C converted transcripts (adjusted p-value < 0.05) between KLF5 degradation (dTAG^v-1 treatment) and control by significance and selected the top downregulated genes. Only genes that were downregulated in each replicate analyzed were kept. Each gene set was further stratified to include only genes with a KLF5 ChIP-seq peak called by MACS2[90] and annotated by HOMER[91] at or adjacent to that gene. To generate KLF5 RNA-seq targets, we compiled the 250 most significantly downregulated genes following KLF5 KO in each cell line. To generate the Classical KLF5 Peaks and Basal-like KLF5 peaks gene sets, we compiled all protein-coding genes annotated by HOMER in each respective KLF5 ChIP-seq dataset. GSEA was performed using rank-ordered log₂(Fold Change) values and the GSEA "Preranked" option. Gene Sets for GSEA are included as Supplementary Data 11.

Transcriptomic analysis of patient samples was performed using normalized count data (Bailey et al.[11]) or raw sequencing reads deposited to GSE93326[20], EGAD00001003582[54], or EGAD00001005799[13]. Raw sequencing reads were first aligned to the hg38 genome using STAR v2.7.9. Samples were stratified according to the published tumor classifications. Transcriptomic analysis of mouse single-cell RNA-sequencing data was performed by first aligning raw sequencing reads deposited to GSE207943[55] to the mm10 genome using STAR v2.7.9. TPMs were calculated for each set of samples as described above.

Transcriptomic analysis of human cancer cell lines was performed using normalized count data extracted from the Cancer Cell Line Encyclopedia (CCLE) (DepMap 24Q4). A Basal-Classical gene score for each PDAC cell line was calculated by first generating z-scores for classical and basal-like PDAC based on the genes defining each subtype in Moffitt et al.[12], Bailey et al.[11], and Chan Seng Yue et al.[13], then subtracting the 3 classical z-scores from the 3 basal-like z-scores.

### SLAM-seq analysis
GENCODE v44 hg38 3'UTR exons were downloaded as a BED file from the UCSC Genome Browser (https://genome.ucsc.edu/cgi-bin/hgTables). Single-end 100 base pair sequencing reads were mapped to hg38 3'UTR exons, filtered, T→C snps identified, and reads counted using the slamdunk[92] package with default settings. Median conversion rates were calculated using the alleyoop utrrates option within the slamdunk package. Counted T→C converted transcript reads were compiled for each gene in each sample by summing reads mapping to different 3'UTR exons for a given gene. Low abundance transcripts were removed, and variance stabilized normalized transcripts and differential expression were calculated using DESeq2[87]. Analyses of processed SLAM-seq data were performed as described for RNA-seq data.

### ChIP-seq analysis
Single-end 76 or 100 base pair sequencing reads were mapped to the hg38 genome using Bowtie2[93] with default settings. MACS2 v2.2.9.1[90] was used to call peaks using input genomic DNA as the control. Annotation and motif analysis of ChIP-seq peaks was performed using HOMER v5.11 with default settings[91]. To visualize genomic tracks, bigWig files were generated from sorted, indexed BAM files using deepTools[94] v3.5.2 bamCoverage function. Reads from single-end sequencing were extended based on sonication fragment size (300 base pairs). For experiments involving KLF5 degradation or RUVBL1/2 inhibition, sequencing reads were first mapped to the mm10 genome. Unaligned reads were then mapped to hg38 for downstream analysis. bigWig files were normalized according to the number of reads aligned to the mouse genome for each sample.

To define BED files of peaks and peak overlaps, MACS2 output narrowPeak or broadPeak files were merged using bedtools[95] v2.30.0 intersect tools. Regions of high artifactual mapping to chromatin, "Blacklisted" regions, were removed from BED files using bedtools intersect prior to each analysis[96]. Heatmaps and average chromatin occupancy metaplots were generated using computeMatrix and plotHeatmap functions of deepTools, taking bigWig and BED files as input.

### Single-nucleus RNA-sequencing analysis
Analysis was performed in the same manner as described in Hwang et al.[56]. Briefly, CellRanger aligned-CellBender corrected FASTQ files were downloaded from GSE202051. Nuclei with over 500 UMI counts were then filtered and normalized by total counts over all genes, followed by log2(X + 1) as the final expression unit. All patients were aggregated into a single dataset. The log1p_norm expression matrix was constructed for downstream analyses. The Scanpy[97] 1.7.2 highly_variable_genes function was used with sample ID as input for the batch to identify the top 4000 highly variable genes across the dataset, upon which a principal-component analysis was performed to identify the top 30 principal components beyond which negligible additional variance was explained in the data. Batch correction was not applied. Individual nucleus profiles were visualized using UMAP, with individual nuclei plotted as circles in the UMAP. Scrublet[98] v0.2.3 was used to identify and remove doublets. Distinct cell populations identified from the previous steps were annotated using known cell-type-specific gene expression signatures and representative gene markers. The AMI score was computed using the adjusted_mutual_info_score function in the scikit-learn[99] v0.22.2 package, then was used to quantify similarity between single-cell assignments between the partitions imposed by the Leiden clustering labels and patient ID labels. We filtered out malignant cells with less than 500 counts and genes expressed in at least 10 malignant cells. To score gene signatures for each nucleus profile, a signature score for each nucleus profile was computed as the mean log1p_norm expression across all genes in the signature. For this analysis, we generated Moffitt Classical and Moffitt Basal-like signatures as explained above, filtering to a maximum of 75 genes in each dataset.

### Analysis of publicly available data
Cancer dependency map (DepMap[61]) gene essentiality 24Q1 data was downloaded in April 2024.

### Cloning and molecular biology
Oligonucleotides for primers and sgRNAs were ordered from Sigma-Aldrich. Pre-made vectors and gene cassettes were ordered from Addgene, and new DNA fragments were ordered as gBlocks from IDT. The Takara® In-Fusion HD or Snap Assembly cloning kit was used to clone new plasmids according to the manufacturer's instructions. Plasmids were transformed into Stbl3™ competent E. coli for antibiotic selection and plasmid amplification. All plasmid sequences were sequenced for validation prior to use. All plasmids have since been subject to whole plasmid sequencing (Plasmidsaurus) for additional validation. Plasmids and cDNA identities are included as Supplementary Data 12.

### CRISPR-based targeting for competition dependency assays
For GFP-depletion assays, cells stably expressing Cas9 from a LentiV-Cas9-blasticidin vector were lentivirally transduced with sgRNAs in an LRG2.1_puromycin[80] vector (Addgene # 125594) at a ~30–50% GFP-positivity rate. GFP percentage was measured using a MilliporeSigma®

Guava easyCyte Flow Cytometer on day three post viral transduction, then once every three days for 21 days. Two measurements were made for GFP% of each well, and the measurements were averaged at each time point. All competition data was collected in duplicate or triplicate. Oligonucleotide sequences are included as Supplementary Table 2.

## Gene complementation assays

*KLF5* wildtype (WT) and mutant cDNA constructs were cloned into a lentiviral FLAG fusion expression vector containing a neomycin resistance cassette, which was used to produce lentiviral supernatant. Rescue constructs, or an empty vector control, were stably expressed in AsPC-1 cells, and infected cells were selected with neomycin (G418) for 7–10 days. Expression of each cDNA was validated by FLAG Western blot before proceeding. cDNA-expressing cells were infected with lentiviral sgRNA targeting *KLF5* (2 sgRNAs) or the *Rosa26* safe harbor locus (control) at a ~30–50% GFP-positivity rate. The percentage of GFP-expressing cells in the population was measured every three days for 21 days using a MilliporeSigma® Guava easyCyte Flow Cytometer. Rescue percentage was calculated by normalizing the percentage of GFP-expressing cells on day 15, relative to day 3, for each rescue construct to the percentage of GFP-expressing cells on day 15, relative to day 3, for WT KLF5.

## Bacterial expression and purification of recombinant MBP, MBP-KLF5, RUVBL1, RUVL2, and RUVBL1/2

6xHis-TEV-MBP, 6xHis-TEV-MBP-KLF5, and 6xHis-GFP-TEV-RUVBL2 expression was induced in BL21-CodonPlus (DE3)-RIPL (Agilent) cells grown in Luria broth (LB) supplemented with antibiotics and 1 mM Isopropyl β-D-1-thiogalactopyranoside (IPTG) for 18 h at 16 °C. Additionally, 6xHis-TEV-RUVBL1 expression was co-induced with untagged-RUVBL2 in BL21 cells. Cells were collected by centrifugation at 6000 RCF for 20 min at 4 °C, and resuspended in 30 mL lysis buffer per 1 L of liquid culture (50 mM $Na_3PO_4$, pH 7.5, 500 mM NaCl, 0.1 mM EDTA, 1% Triton X-100, 10 mM imidazole, 5% glycerol, 1 mM DTT (fresh)), supplemented with protease inhibitors and 100 µg/mL lysozyme. Cells were sonicated for 2 min and 30 s (2 s on, 4 s off) with a probe sonicator at 40% amplitude. Lysates were clarified by ultracentrifugation at 25,000 RCF for 1 h at 4 °C. The soluble supernatant was loaded onto an affinity column containing 500 µl Ni-NTA resin/1 L liquid culture (Qiagen), pre-equilibrated with lysis buffer. After the supernatant was allowed to flow through, the column was washed with 3x with 15 mL wash buffer (20 mM $Na_3PO_4$, pH 7.5, 500 mM NaCl, 30 mM imidazole, 0.1% β-mercaptoethanol, supplemented with protease inhibitors). Target proteins were eluted in 3 fractions of 1 mL each of elution buffer (20 mM $Na_3PO_4$, pH 7.5, 500 mM NaCl, 200 mM imidazole, 0.5 mM EDTA, 1 mM DTT (fresh), supplemented with protease inhibitors). TEV cleavage was achieved by diluting 3 mL eluent to 10 mL in dilution buffer (20 mM $Na_3PO_4$, pH 7.5, 0.5 mM EDTA, 1 mM DTT (fresh), supplemented with protease inhibitors), followed by the addition of TEV protease (NEB #P8112) to a final concentration of 15 µg/mL and overnight incubation. TEV protease and cleaved 6x-His tags were removed by flow through an affinity column loaded with equilibrated Ni-NTA resin, with the flow-through containing the target proteins.

For purification of MBP- and MBP-KLF5 for use in mass spec experiments, eluted protein was incubated with 2 mL of amylose resin per 1 L of liquid culture (NEB #E8021L) for 1–2 h. Amylose resin was pre-washed 3x in 10 mL wash buffer (50 mM $Na_3PO_4$, pH 7.5, 0.1 mM EDTA, 500 mM NaCl, 1 mM DTT, 5% Glycerol). After incubation, the resin was pelleted by centrifugation for 5 min at 1000 RCF, washed 3x in 12 mL wash buffer, and resuspended in 3 mL elution buffer (50 mM $Na_3PO_4$, pH 7.5, 0.1 mM EDTA, 150 mM NaCl, 1 mM DTT, 15 mM maltose, 5% Glycerol). The resin was rotated during elution buffer incubation for 10 min, then pelleted by centrifugation for 5 min at 1000 RCF. Supernatant was collected as elution sample, three total rounds of elution

were performed, and the eluents were pooled. Protein concentration was measured by NanoDrop.

A size exclusion purification step followed each purification using an AKTA® Pure 25 M (Cytiva 29018226) using a Superdex® 200 Increase 10/300 GL (for MBP, MBP-KLF5, and RUVBL2) or a Superose® 6 Increase 10/300 GL (for RUVBL1/2) with SEC running buffer (50 mM $Na_3PO_4$ pH 7.5, 0.1 mM EDTA, 200 mM NaCl, 1 mM DTT (fresh), 5% glycerol). Monomeric RUVBL1 unbound to RUVBL2 was separately collected by SEC following the RUVBL1/2 purification. Purity was evaluated by SDS-PAGE and Coomassie staining, and protein used immediately, or flash frozen in liquid nitrogen and kept at −80 °C. All purified proteins were validated by mass spectrometry peptide identification and Western blot. All steps were performed at 4 °C.

## KLF5-RUVBL1/2 direct interaction co-immunoprecipitation assays

5 µg of purified MBP or 10 µg of recombinant MBP-KLF5 not previously subject to MBP purification were added to 250 µL Amylose resin per IP. Proteins were allowed to incubate by rotating for 30 min. The resin was pelleted by centrifugation for 5 min at 1000 RCF then washed 2x in 1 mL wash buffer (50 mM $Na_3PO_4$ 7.5, 0.1 mM EDTA, 250 mM NaCl, 1 mM DTT, 5% Glycerol), after which 10 µg of RUVBL1, 10 µg of RUVBL2, or 20 µg of the RUVBL1/2 hetero-hexamer was added to each IP and incubated with MBP-bound Amylose resin in resin binding buffer (50 mM $Na_3PO_4$ pH 7.5, 0.1 mM EDTA, 200 mM NaCl, 1 mM DTT(fresh), 5% glycerol) by rotation for 2 h. Each IP was then pelleted by centrifugation, washed 2x in 1 mL wash buffer, and eluted by incubation for 10 min in 250 µL elution buffer (50 mM $Na_3PO_4$, pH 7.5, 0.1 mM EDTA, 200 mM NaCl, 1 mM DTT, 15 mM maltose, 5% Glycerol). Two rounds of elution were performed for each IP. Supernatant containing the eluted proteins was then diluted 1:1 in 2x Laemmli buffer with 5% β-mercaptoethanol, boiled for 15 min at 98 °C and run on an SDS-PAGE gel for Western blotting. Apart from sample boiling, all steps were performed at 4 °C.

## Nuclear extraction

Cells were harvested by trypsinization and washed 1x with ice-cold PBS. Per 100 million cells, cells were then incubated in 10 mL cytoplasmic extraction buffer (20 mM Tris pH 7.5, 10 mM KCl, 1.5 mM $MgCl_2$, 340 mM sucrose, 10% glycerol, 0.1% Triton X-100, 1 mM DTT, supplemented with protease inhibitors) for 10 min. Nuclei were pelleted for 5 min at 1000 RCF, washed 3x in 10 mL cytoplasmic extraction buffer, and resuspended in 1 mL nuclear extraction buffer (20 mM Tris pH 7.5, 1.5 mM $MgCl_2$, 0.5% Triton X-100, 5% Glycerol, 200 mM NaCl, 1 mM DTT, supplemented with protease inhibitors). Samples were rotated for 2–4 h, then nuclear lysates were cleared by centrifugation at 20,000 RCF for 30 min. For IP-Mass Spec experiments, chromatin was digested by the addition of 1 µL Benzonase® Nuclease (MilliporeSigma #E1014) per 1 mL nuclear extraction buffer and incubation for 1 h prior to clearing by centrifugation. Nuclear extract supernatant was then collected for analysis or follow-up experiments. All steps were performed at 4 °C.

## Immunoprecipitation mass spectrometry

For recombinant KLF5 immunoprecipitation coupled with mass spectrometry (IP-MS), 30 µL of magnetic protein G beads per IP were washed 2x in 1 mL bead binding buffer (50 mM Tris pH 7.5, 150 mM NaCl, 0.1 mM EDTA, 5% Glycerol, 1 mM DTT, supplemented with protease inhibitors), followed by overnight incubation with 7.5 µg of MBP antibody in 250 µL of bead binding buffer by rotating. Beads were then washed 3x in 500 µL of bead binding buffer, then 5 µg purified MBP- or MBP-KLF5 was added to each IP and incubated for 4 h in 250 µL of bead binding buffer by rotating. Beads were then washed 3x in 500 µL of bead binding buffer, then nuclear extract was pre-cleared by incubation for 1 h with magnetic protein G beads bound to MBP antibody and

recombinant MBP protein. For each IP, 800–1500 μL of cleared nuclear extract (80–150 million cells) was added and incubated by rotating for 1–2 h. Beads were then washed 5x in 250 μL bead washing buffer (20 mM Tris pH 7.5, 1.5 mM $MgCl_2$, 5% Glycerol, 120 mM NaCl, 1 mM DTT) and beads were resuspended in 40 μL 2x Laemmli buffer with 5% β-mercaptoethanol. Samples were boiled for 15 min at 98 °C and run on an SDS-PAGE gel for gel-digestion, followed by analysis by MS. All steps were performed at 4 °C, and three biological replicates were included for each condition for all IP-MS experiments.

IPs were then subjected to analysis by Liquid chromatography-mass spectrometry (LC-MS) at the Cold Spring Harbor Mass Spectrometry Core Facility (Cold Spring Harbor, NY 11724). Briefly, peptides were first loaded via 10 cm × 100 μm ID trap column packed with 5 μm Aqua C18 particles (Phenomenex) and separated on a 30 cm × 75 μm ID analytical column packed with 1.9 μm Reprosil C18 silica. Chromatographic separation was performed using a 5–35% acetonitrile gradient in water (0.1% formic acid) at a 200 nL/min flow rate. Eluting peptides were then ionized by electrospray at 2200 V and transferred into an Orbitrap Fusion Lumos Tribrid mass spectrometer (Thermo). Precursor MS scans were collected in the Orbitrap at a resolution of 120,000 across an m/z range of 380–2000 Th. Selected precursor ions were fragmented using higher-energy collisional dissociation (HCD) with stepped normalized collision energies of 30, 35, and 40% (data-dependent mode). The first mass was set to 100 Th, and fragment spectra were collected in the ion trap at normal scan rates.

Raw data were processed using ProteomeDiscoverer with the Mascot scoring function. Mass tolerances were set to 5 ppm for MS MS1 and 0.5 Da for MS2. Spectra were searched specifically against the *Escherichia coli* (strain K12) protein database and human protein databases, as well as a common contaminant database (cRAP). For this search, variable modifications included M-oxidation and N/Q deamidation. Percolator was used to filter peptide spectral matches to maintain a false discovery rate (FDR) < 0.01 (1%). Label-free quantification (LFQ) values were determined by integrating precursor ion intensities from the extracted chromatographs, and LFQ was used to quantify relative protein abundance.

The mass spectrometry data have been deposited to ProteomeXchange via the PRIDE repository (dataset identifier PXD066112).

### RUVBL1/2 co-IPs
1 mL (100 million cells) nuclear extracts from cells stably expressing FLAG-tagged RUVBL1 per IP were incubated with 25 μL agarose M2 FLAG® (MilliporeSigma A2220) beads overnight on a rotator. FLAG beads were washed 2x in 1 mL nuclear extraction buffer (20 mM Tris pH 7.5, 1.5 mM $MgCl_2$, 0.5% Triton X-100, 5% Glycerol, 200 mM NaCl, 1 mM DTT, supplemented with protease inhibitors) and 2x in 1 mL FLAG washing buffer (50 mM Tris pH 7.5, 150 mM NaCl, 0.5 mM EDTA, 10% glycerol, 1 mM DTT). After the last wash, samples were eluted by incubation with 3X FLAG® peptide (MilliporeSigma #F4799) at 200 μg/mL in 40 μL FLAG washing buffer. Two rounds of elution were performed. Elution fractions were then diluted 1:1 in 2x Laemmli buffer with 5% β-mercaptoethanol. Samples were boiled for 15 min at 98 °C and run on an SDS-PAGE gel for gel-digestion, followed by analysis by MS. 1 μM CB-6644 or DMSO was added to each of the buffers in the IP.

### HEK293T co-immunoprecipitation assays
HEK293T cells were first transfected with 10 μg of FLAG-tagged cDNA plasmid using polyethyleneimine. For RUVBL1/2 co-IP experiments, HEK293T cells were also simultaneously transfected with 3 μg of N-terminal HA-tagged RUVBL1 and 3 μg of C-terminal HA-tagged RUVBL2 cDNA plasmids. Transfection media was changed 6 h following transfection. Forty-eight hours after transfection, HEK293T cells were collected by scraping and washed 1x with ice-cold PBS. Cells were then incubated for 10 min with 2 mL of cell lysis buffer (20 mM Tris pH 8.0, 1.5 mM $MgCl_2$, 10 mM KCl, 15% glycerol, 1 mM DTT) supplemented

with protease inhibitors per 10 cm dish of cells. Nuclei were pelleted for 5 min at 600 RCF and resuspended in 1.3 mL nuclear extraction buffer (20 mM Tris pH 8.0, 1.5 mM $MgCl_2$, 210 mM NaCl, 15% glycerol, 0.2 mM EDTA, pH 8.0, 0.1% Tween-20, 1.5 mM $MgSO_4$) supplemented with protease inhibitors. Samples were rotated at 4 °C for 1 h, then nuclear lysates were cleared by centrifugation at 20,000 RCF for 30 min and subsequently incubated with 25 μL agarose M2 FLAG® (MilliporeSigma #A2220) beads overnight on a rotator. FLAG beads were washed 3x in nuclear extraction buffer and pelleted at 8200 RCF for 30 s prior to adding cleared nuclear lysates. After overnight incubation, lysates were washed 1x with 1 mL of nuclear extraction, followed by 3x washes with wash buffer (50 mM Tris pH 7.5, 150–350 mM NaCl, 0.5 mM EDTA, 10% glycerol). After the last wash, IP beads were pelleted at 8200 RCF for 30 s and resuspended in 35 μL 3x Laemmli buffer with 5% β-mercaptoethanol. Samples were boiled for 15 min at 98 °C and run on an SDS-PAGE gel for Western blotting. Samples were normalized for equal concentration by BCA of the Input for each IP. Apart from sample boiling, all steps were performed at 4 °C. For co-immunoprecipitation experiments involving CB-6644, HEK293T cells were treated with CB-6644 for 4 h prior to IP. CB-6644 was also added to lysis and wash buffers.

### Western blots and protein analysis
Cells were counted for each sample and normalized for cell number. Equal numbers of cells were washed in PBS and resuspended in 1x Laemmli buffer, diluted in PBS, with 5% β-mercaptoethanol at a final concentration of $5 \times 10^6$ cells/mL. Samples were boiled at 98 °C for 15 min and stored at 4 °C for a maximum of one month, or at −20 °C for long-term storage. Immediately prior to loading on an SDS-PAGE gel, samples were boiled for 1 min and centrifuged at 20,000 RCF for 5 min. Antibodies are included as Supplementary Table 3.

### CellTiter-Glo
Cells were grown in opaque white 96-well plates for the duration of the assay. At the endpoint, CellTiter-Glo® reagent was added at a buffer: media ratio of 1:2 and incubated at room temperature for 10 min on a shaker at low speed. Luminescence was subsequently measured using a Molecular Devices SpectraMax i3.

### RUVBL1/2 inhibitor-resistant mutant experiments
*RUVBL1* or *RUVBL2* wildtype (WT) and mutant cDNA constructs were cloned into a lentiviral expression vector containing a neomycin resistance cassette, which was used to produce lentiviral supernatant. cDNA constructs, or an empty vector control for RNA-seq, were stably expressed in AsPC-1, T3M-4, or SUIT-2 cells, and infected cells were selected with G418 for 7 days. cDNA-expressing cells were then treated with CB-6644 and subject to analysis by CellTiter-Glo or RNA-seq.

### Animal studies
All animal procedures and studies were approved by the Institute of Animal Care and Use Committee of CSHL and were conducted in accordance with the National Institutes of Health (NIH) Guide for the Care and Use of Laboratory Animals. For all animal studies, mice of similar age and gender were co-housed and block randomized in an unblinded manner. Female mice ranging from 7 weeks old to 12 weeks old were utilized in the described experiments, and mice were age-matched with appropriate control mice for analysis. All mice were euthanized using $CO_2$ inhalation following institutional guidelines. NSG (NOD.Cg-Prkdc scid Il2rg tm1Wjl /SzJ, Strain # 005557; RRID:IMSR_JAX:005557) mice used for the PDAC orthotopic transplantation studies were purchased from The Jackson Laboratory. All mice were housed in specific pathogen-free facilities at CSHL under a 12 h:12 h light/dark cycle, with food and water available *ad libitum*. All purchased mice were acclimated to their facility for at least one week prior to enrollment in experiments.

## Mouse experiments

To perform the primary pancreatic tumor model, cancer cells were orthotopically transplanted into the pancreas as previously described[52]. Briefly, mice were anesthetized using continuous isoflurane, and analgesic (2 mg/kg meloxicam and 5 mg/kg bupivacaine) was administered preemptively. The fur of the left abdominal flank was cut using veterinary clippers and then treated using depilatory cream. Depth of anesthesia was assessed prior to cleaning and disinfecting the skin and making a 1 cm incision to both the skin and underlying peritoneum to access the peritoneal cavity. The pancreas was exteriorized, and a 30-gauge needle was used to inject 100,000 luciferase-expressing AsPC-1 or T3M-4 cells two days following lentiviral delivery of *sgRNAs* to knockout *KLF5* or control. Cells were suspended in 20 μL of a 1:1 mixture of 1x DPBS and growth factor-reduced Matrigel [Corning Cat#356231] and injected into the tail of the pancreas. Successful injection was confirmed by the formation of a liquid bleb at the site of injection with minimal fluid leakage. The pancreas was then returned to the peritoneal cavity, and the peritoneum and skin were closed using 4−0 Vicryl suture and wound clips, respectively. All mice were monitored by bioluminescence every 7 days for 7−8 weeks, as described below. Over the course of primary tumor growth, mice were monitored 3−4 times per week for general health and euthanized early based on defined humane endpoint criteria according to our animal protocol, including tumor diameter ≥2 cm, ascites, jaundice, lethargy, ≥20% body weight loss, or other signs of sickness or distress. Maximum tumor size of ≥2 cm (diameter) was not exceeded.

To perform luciferase imaging, mice were first intraperitoneally (IP) injected with 100 μL of luciferin (50 mg/kg) into the lower right quadrant with a 27 G needle (BD Biosciences, 30519). Mice were then anesthetized with isoflurane, and bioluminescence was imaged with an IVIS Spectrum (Xenogen) 15 min following IP injection. Ventral, dorsal, and both flanks were imaged for each mouse at each timepoint, and the sum of the four measurements was recorded as the bioluminescence for that timepoint.

To perform endpoint tumor analysis, tumors were resected at endpoint and mechanically minced using parallel razor blades. For protein analysis, tumor samples were resuspended in 400 μL RIPA buffer, followed by 10 x 30 s high-frequency sonication, resuspension in 2x Laemmli buffer with 5% β-mercaptoethanol, and boiling for 15 min at 98 °C. Western blotting was performed as described above. For RNA analysis, tumor samples were resuspended in 1 mL TRIzol reagent. RNA extraction and RT-qPCR were performed as described above. For genomic DNA analysis, tumor samples were first resuspended in 400 μL DNA extraction buffer (10 mM Tris-HCl, pH 8.0, 150 mM NaCl, and 10 mM EDTA), supplemented with 4 μL 10% SDS and 4 μL Proteinase-K, followed by 24-h incubation at 54 °C. Genomic DNA was extracted using a Quick-DNA Genomic DNA Purification Kit (ZYMO Research). Select genomic regions were amplified by PCR (primer sequences in Supplementary Table 1) and sent for Sanger sequencing (Eurofins Genomics, Louisville, KY 40299).

## Statistics and reproducibility

All statistical tests used to evaluate significance are detailed in the respective figure legends. All reported results were replicated across multiple experiments to generate reliable results, and replication is described in more detail in each figure legend. No statistical method was used to predetermine sample size. Sample size calculations were based on previously published data. Figure legends indicate the sample sizes for each set of experiments. All sample sizes were determined to be sufficient given that the differences among groups were consistent. No data were excluded from any experiment except for the knockout RNA-seq analysis in T3M-4 (Fig. 3G, H). For this experiment, one *RUVBL1* knockout sample was excluded from analysis due to low library concentration resulting from ineffective library preparation. For the orthotopic PDAC experiments, age-matched mice were

randomly assigned to control or *sgKLF5* groups. Orthotopic transplantations for control and treatment groups were performed on the same day. Randomization was not performed for any other experiment. Investigators were not blinded to allocation during experiments and outcome assessment.

## Reporting summary

Further information on research design is available in the Nature Portfolio Reporting Summary linked to this article.

## Data availability

All genomic datasets are available at the GEO database under accession codes GSE295347 (SLAM-seq), GSE295348 (CRISPR screens), GSE295349 (RNA-seq), and GSE295354 (ChIP-seq). The mass spectrometry data have been deposited to the ProteomeXchange Consortium via the PRIDE partner repository with the dataset identifier PXD066112. The Moffitt et al. human PDAC dataset was acquired from GSE71729. The Aung et al. and Chan Seng Yue et al. human PDAC datasets were acquired from EGAS00001002543. The Maurer et al. human PDAC dataset was acquired from GSE93326. The Burdziak et al. mouse PDAC progression scRNA-seq dataset was acquired from GSE207943. The Hwang et al. human PDAC snRNA-seq dataset was acquired from GSE202051. All source data and raw uncropped Western Blots are published alongside this paper as a Source Data file. Unnormalized and unprocessed source data are available at Figshare (https://doi.org/10.6084/m9.figshare.c.8082880). Source data are provided with this paper.

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

## Acknowledgements

This study was supported by the Cold Spring Harbor Laboratory NCI Cancer Center Support under grant CA045508. C.R.V. was supported by the Pershing Square Sohn Cancer Research Alliance, NCI grants CA013106-Project 1, CA229699, and CA290004. Additional funding was provided by Cold Spring Harbor Laboratory and Northwell Health Affiliation and Treeline Biosciences. P.J.C. was supported by NCI grant CA278591. N.S. was supported by NCI grant CA281246. D.S. was supported by NCI R50 grant CA305054.

## Author contributions

P.J.C. and C.R.V. conceived this project, designed the experiments and wrote the paper with input from all of the authors. P.J.C. performed the experiments and analyzed the data with the following support: N.S. performed orthotopic transplantations and assisted with mouse experiments, and M.E. provided advice; D.S. performed FACS and assisted with tissue culture for the reporter screens; O.K. analyzed the human PDAC sequencing datasets, including the snRNA-seq data; D.T. performed the HNF4α, p63, and dTAG$^v$-1 T3M-4 H3K27ac ChIP and dTAG$^v$-1 competition assays. D.M.-S. provided methodology for the reporter screening. C.R.V. acquired the funding and supervised the studies.

## Competing interests

C.R.V. has received consulting fees from Flare Therapeutics, Roivant Sciences and C4 Therapeutics; has served on the advisory boards of KSQ Therapeutics, Syros Pharmaceuticals and Treeline Biosciences; has received research funding from Boehringer-Ingelheim and Treeline Biosciences; and owns a stock option from Treeline Biosciences. M.E. owns stock in Agios Pharmaceuticals. The remaining authors declare no competing interests.
