## [Transparent Peer Review file · Nature Communications]

KLF5 drives dichotomous lineage programs in pancreatic cancer via the AAA+ ATPase coactivators RUVBL1 and RUVBL2

Corresponding Author: Professor Christopher Vakoc

Version 0:

Reviewer comments:

Reviewer #1

(Remarks to the Author)

The study by Cunniff et al. reports that KLF5 is highly expressed in pancreatic ductal adenocarcinoma (PDAC) and plays a key role in regulating classical and basal lineage identity. Mechanistically, the authors demonstrate that KLF5 interacts with transcriptional coactivators RUVBL1/2 via its disordered region to modulate cis-regulatory elements in PDAC cells. Importantly, they show that chemical inhibition of RUVBL1/2 suppresses KLF5 function and lineage identity genes. This work is compelling, well-executed, and merits publication with minor revisions to clarify certain points and expand discussions.

Major Points for Clarification/Discussion:

1. KLF5 and H3K27ac Regulation: The authors frequently use H3K27ac ChIP-seq as a readout for cis-regulatory elements. Could they discuss whether KLF5 regulates H3K27ac levels directly (e.g., through TIP60/P400) or indirectly? Alternatively, could this be further explored by analyzing enhancer RNAs (eRNAs) via SLAM-seq?
2. KLF5-RUVBL1/2 Axis in Cancer Biology: The interaction between KLF5 and RUVBL1/2 is intriguing. Could the authors speculate whether this axis plays a role in other cancers? For instance, in other malignancies, might RUVBL1/2 cooperate with different lineage-specific transcription factors? Additionally, is this mechanism specific to cancer progression, or could it also have developmental implications?
3. Evolutionary and Mechanistic Insights: The analogy between NtrC1 AAA+ ATPase and RUVBL1/2 in transcriptional activation provides valuable evolutionary context. Could the authors elaborate further in the discussion (or even abstract) on the conserved nature of this regulatory cascade? Additional analyses of RUVBL1/2 mass spectrometry data in specific cell types could further strengthen this argument.

Minor Points:

1. Clarification on RUVBL1/2 ATPase Activity: The statement that "ATPase activity of RUVBL1/2 is required for KLF5 binding" could be misinterpreted as affecting KLF5's chromatin binding. It may be clearer to state that "ATPase activity of RUVBL1/2 is required for its association with KLF5."
2. SLAM-seq Spike-in Controls: Did the authors use spike-in controls for SLAM-seq normalization? If not, this should be noted in the methods or discussed, especially if global transcription changes could affect data interpretation.

Reviewer #2

(Remarks to the Author)

In this manuscript, Cunniff et al present a comprehensive study combining multiple genomic approaches, including reanalysis of published data, 4sU-labeling RNA-seq, genome-wide CRISPR screening, protein interactome analysis, and in vivo transplantation assays, to identify KLF5 and its interaction with RUVBL1/2 in regulating epithelial feature genes in both classical and basal-like pancreatic ductal adenocarcinoma. The study is data-rich and technically impressive, generating

valuable insights into the molecular mechanisms driving cancer cell proliferation. However, the logical flow and clarity of presentation could be improved to better highlight the novel findings and strengthen the overall narrative. Below are my concerns.

Major:

1. The study's logical flow begins with a reanalysis of a previously published CRISPR screen dataset from the same research group. In addition to the already reported Np63, the authors identified KLF5 as another key factor that shows high expression in both basal-like and classical PDAC subtypes. To investigate KLF5's functional role, the authors employed a degradation system targeting KLF5 in cell lines, coupled with 4sU-labeling SLAM-seq analysis. This approach revealed SERPINB5 as one of the common differentially expressed genes downstream of KLF5. The authors then combined additional CRISPR screening with SERPINB5-based FACS sorting, ultimately identifying RUVBL1/2 through subsequent immunoprecipitation experiments.

Within this logical framework, SERPINB5 emerges as what should be a novel and central player connecting KLF5 to RUVBL1/2 in the proposed regulatory pathway. However, despite this apparent importance, the study provides surprisingly limited experimental validation and discussion of SERPINB5's specific role. This relative neglect of SERPINB5 characterization creates a noticeable gap in the otherwise logical progression from KLF5 to RUVBL1/2, making it challenging to fully appreciate how these components functionally interconnect. The manuscript would benefit significantly from more thorough investigation and discussion of SERPINB5's precise mechanistic role in bridging KLF5's transcriptional regulation with the RUVBL1/2 complex.

Specifically:

a) Throughout the manuscript, including the abstract, introduction, and discussion sections, there is remarkably little discussion about the biological function of SERPINB5 in PDAC, nor its functional relationship with the identified KLF5 and RUVBL1/2 components.

b) The data presented in Figure 3F reveal an interesting asymmetry: while knockout of KLF5 significantly affects SERPINB5 expression, knockout of SERPINB5 has no apparent effect on KLF5 levels. This observation raises important questions about whether SERPINB5 expression can truly serve as a reliable indicator of KLF5 activity. If SERPINB5 expression doesn't consistently reflect KLF5 status, this potentially undermines the validity of using SERPINB5 antibody-based FACS sorting as the screening method in the genome-wide CRISPR screen shown in Figure 3C. Since the authors used SERPINB5 expression to identify RUVBL1/2 as key interactors, it becomes critically important to determine whether the functional interaction between KLF5 and RUVBL1/2 actually depends on SERPINB5. To address this question, the authors should perform SERPINB5 knockout experiments to verify whether RUVBL1/2's binding to KLF5 requires SERPINB5's presence.

c) While the SLAM-seq data presented in Figures 2C-D and 5I-J (measuring newly synthesized RNA) theoretically provide advantages over total mRNA analysis by distinguishing transcriptional effects from post-transcriptional regulation, the manuscript would benefit from directly comparing these results with using total RNA expression data from the same samples. Specifically, it would be particularly informative to see whether the key findings regarding SERPINB5 hold true in only newly synthesized rather than total RNA analyses, as one would expect the most robust hits (like SERPINB5) to be only identified in new RNA datasets. Additionally, to properly validate the SLAM-seq methodology, the authors should include quality control metrics in at least the supplementary materials. These should include data on the efficiency of 4sU labeling and chemical conversion, such as the T→C mutation rates and the proportion of newly synthesized RNA in each sample, as well as controls without chemically induced mutation. Such QCs are essential for readers to evaluate the robustness and validity of the SLAM-seq analysis.

2. The study presents comprehensive datasets that are both data-intensive and scientifically substantial. However, the current visualization of these results in the figures presents significant challenges for reader comprehension. The graphical representations could be substantially improved to enhance clarity and facilitate easier interpretation of the important findings. Several specific issues with the figure presentations should be addressed. For example, multiple figures lack essential labeling information that is crucial for proper interpretation. For example: Figures 2C and 2D: The x-axis label "log2(fold change)" in these figures lacks explanation of what specific comparison is being shown. Figures 2H and 2I similarly lack y-axis labels. Key axis information is also missing in Figures 3B, 3G, 5F and 5K. These labeling deficiencies extend to several supplementary figures as well. Additionally, the relationship between the three heatmaps presented in Figure 4H requires clarification. It remains unclear whether all three heatmaps represent the identical set of 9,274 peaks or if there are differences between them.

3. In figs. 4D-4F, the IP experiments identify IDR2 as the essential domain for KLF5-RUVBL1/2 interaction. However, Figure 4F shows that the depletion of other domains (DBD and IDR) displays similar interaction patterns to IDR2 deletion. This suggests the possibility that KLF5's function may not rely solely on RUVBL1/2 binding through IDR2, and that additional mechanisms could be involved? The authors should clarify and discuss this observation. Additionally, why does KLF5 over-expression fail to completely restore function to 100%?

Minor:

Line 69: Remove duplicate "cell lines"

Fig. 3C: Define "corepressors" and "coactivators"

Fig. 3H: Clarify CDK1's role

Standardize nomenclature: "RPA3" (Fig. 3H) vs. "RPAP3" (Line 235)

Fig. 5G: Explain what CB-644 and dTAG-1 are (inhibitors?)

Reviewer #3

(Remarks to the Author)

Dear Editor,

I have reviewed the manuscript entitled "KLF5 drives dichotomous lineage programs in pancreatic cancer via AAA+ ATPase coactivators" by Cunniff, et al. This study focuses on the dichotomous role of KLF5 for both the classical and basal-like subtypes of PDAC. Furthermore, the authors find that the AAA+ ATPases RUVBL1 and RUVBL2 are cooperating with KLF5 across both subtypes as coactivators through interaction with the IDP2 of KLF5. This interaction drives the expression of both key classical and basal genes. RUVBL1 and 2 recruitment at these loci is dependent on KLF5, while KLF5 recruitment itself is not dependent on RUVBL1/2 even though H3K27 acetylation is.

This study has considerable strengths, reflecting both the scope of the work and the rigor of its experimental design. The authors comprehensively examine the role of KLF5 in both the classical and basal-like subtypes of PDAC, revealing how a single transcription factor can drive two distinct transcriptional programs. A key and highly novel finding is that the AAA+ ATPases RUVBL1 and RUVBL2 directly interact with KLF5. For all intents and purposes, the same KLF5-RUVBL1/2 complex is shown to regulate the expression of both classical and basal marker genes. In doing so, the work complements and refines models that are driven by distinct master regulators of PDAC subtypes by identifying a shared TF/co-activator unit that both subtypes utilize. The study shows that once KLF5 is bound, RUVBL1/2 are universally required to establish H3K27 acetylation and drive robust transcription in both contexts. Mechanistically, RUVBL1/2 recruitment to enhancer loci is strictly KLF5-dependent, whereas KLF5 binding occurs independently of RUVBL1/2, establishing a clear hierarchy in complex assembly. By uncovering a common dependency on RUVBL1/2 across PDAC subtypes, the study identifies a unifying therapeutic vulnerability with the potential to overcome subtype-specific resistance mechanisms.

While the study has considerable strengths, several limitations and unanswered questions remain that could benefit from further clarification.

While this work is extensive and methodologically sound in many respects, several areas limit the depth and clarity of its conclusions. The central mechanistic question of how KLF5 distinguishes between classical and basal-specific target genes remains unaddressed, which leaves a gap in understanding the true mechanistic basis of how this TF is functioning and selective of specific basal vs classical genes if in reality it is able to select either. Also, relying exclusively on long-established commercial lines may not fully capture the molecular and phenotypic diversity of these subtypes as they occur in patient tumors, which occur more along a continuum.

The study frames its results as a discrepancy with prior reports citing KLF5 loss in high-grade PDAC or during EMT, yet several other studies have documented high KLF5 expression in PDAC, suggesting the current findings are not necessarily in conflict and could be more accurately positioned within the broader literature.

The results tend to stick to their own lane, in other words, effects are assessed only within each subtype such that basal lines are evaluated for only basal-associated genes, and classical lines for classical-associated genes. Even if the alternate program is minimally or not expressed at baseline, the study should test whether perturbing KLF5 or RUVBL1/2 produces any measurable impact on the other program. Such analyses would clarify whether the proposed shared co-activator module is truly insulated by cell state or has the potential to influence the alternate transcriptional program.

The observation that common KLF5 peaks are predominantly promoter-proximal, while subtype-specific peaks are enriched in intergenic or intronic regions, lacks deeper exploration of the functional implications of this distribution (e.g. enhancers/super-enhancers/chromatin architecture).

Maybe the authors can clarify why in their sgKLF5 RNA-seq raw data that KLF5 itself is not significantly down with other KLFs more significantly impacted in each of the cell lines.

In vivo, KLF5 expression was restored in endpoint tumor samples despite evidence of effective knockout at earlier time points. This raises questions about the cellular source of the recovered signal that are not examined, such as stromal contamination in bulk tissue analysis, repopulation by unedited tumor cells, or selection for resistant/reversion variants. These issues are important to address, as the current data are used to support the conclusion that KLF5 is a "genetic dependency" "in both classical and basal-like PDAC models.

For the mass spectrometry immunoprecipitation, it is unclear which PDAC lysate was used, and because recombinant KLF5 was employed, interactions dependent on endogenous post-translational modifications may have been missed. Although RUVBL1/2 peaks show some subtype specificity, their overall genomic distribution is skewed toward promoters (while the more subtype specific KLF5 peaks skew toward intergenic/intronic), and the BxPC-3 basal-like ChIP-seq dataset appears less convincing for the basal-specific peaks than the classical AsPC-1 dataset for its classical-specific peaks. Given this concern, the authors' conclusion "As observed with KLF5, we found that a significant proportion of RUVBL1/2 peaks were specific to classical or basal PDAC like models" is not fully supported.

Functional assessment of RUVBL1/2 would benefit from chromatin accessibility data (ATAC-seq), which would better capture their direct role in transcriptional regulation than H3K27ac ChIP-seq alone.

Minor comments:

The introduction is quite lengthy, including tangential information (e.g., basal-like PDAC hypersensitivity to KRAS inhibitors) that is not directly relevant to the current work and could be more focused.

Line 51 "progrosis" should presumably be "prognosis"

Antibody information was provided on the reporting summary but would be beneficial to have within the methods or as a supplementary table for the various uses (ChIP-seq versus WB, etc).

Reviewer #4

(Remarks to the Author)

As requested by the editor, this review focuses on the execution, presentation, and interpretation of the pooled CRISPR screens in the manuscript "KLF5 drives dichotomous lineage programs in pancreatic cancer via AAA+ ATPase

coactivators".

The authors performed both genome-wide KO and TF-focused CRISPRi screens in AsPC-1 PDAC cells using an endogenous SERPINB5 (Serpine B5) reporter, binned by flow cytometry into LOW, BULK, and HIGH reporter populations. This approach is appropriate for identifying regulators of the SERPINB5 locus and its upstream transcriptional network.

The genome-wide screen employs the Brunello library, targeting most genes with 4 sgRNAs, which is standard and sufficient for robust identification, whereas the TF screen uses a CRISPRi approach. The reporter screen recovers expected hits such as SERPINB5 (the reporter itself) and KLF5 as the most significantly depleted genes in the Serpin B5-high bin. This demonstrates screen functionality, sensitivity, and specificity. Screens maintain very high coverage for each sgRNA and are sequenced sufficiently deeply. Overall, screen execution, presentation, and interpretation are meticulous, and the screen quality is good.

Raw count tables for each sgRNA across the three groups are provided and easily mapped to gene and group assignments. Three points:

1) The data has several gene names converted to calendar dates, a common error introduced by Excel. This is due to Brunello annotation using outdated HUGO gene IDs, which were updated to avoid this issue. This should be fixed (MARCH1-MARCH11, now called MARCHF1-MARCHF11, SEPT1-12 and SEPT14, now called SEPTIN1-12 and SEPTIN14, respectively, and DEC1, now DELEC1. Please double check that these gene ID's are correct.

2) The TF screen sample "BULK" is strange – "High" and "Low" populations correlate well, but the "BULK" population is uncharacteristically different from both. This indicates to me that something is not quite right with the "Bulk" sample. The authors should look into this again to exclude any errors, and if the poor correlation is indeed in the data, mention this in the text. It would then be best to mainly rely on the "high" to "low" comparison for hit calling. To clarify - this is unlikely to affect any of the conclusions and the identity of the hits, however, should be discussed.

3) More explicit plots or tables showing sgRNA-level variability and consistency for key hits (e.g., KLF5, SERPINB5, RUVBL1/2) would be helpful and demonstrate results more clearly.

In summary, the screens are executed, presented and interpreted correctly, but above issues should be addressed prior to publication.

Minor point:

The method section contains a reference to a genome-wide CRISPRi library, however this was apparently not used? Remove if not needed. Also, throughout the manuscript it would be good to refer to "CRISPR-KO" screen or "CRISPRi screen", to more clearly distinguish the two experiment types.

Version 1:

Reviewer comments:

Reviewer #1

(Remarks to the Author)

The authors have adequately addressed all of my questions. I am happy to recommend this manuscript for publication.

Reviewer #2

(Remarks to the Author)

The authors have successfully address my previous concerns.

Reviewer #3

(Remarks to the Author)

The revised manuscript has adequately addressed reviewer concerns. I do not have any additional comments.

Reviewer #4

(Remarks to the Author)

All points raised have been addressed, I have no further comments.

Manuscript Number: NCOMMS-25-53126-T

Title: KLF5 drives dichotomous lineage programs in pancreatic cancer via AAA+ ATPase coactivators

Response to the Reviewers:

We thank the Reviewers for their thoughtful feedback, suggestions, and critical remarks. We have revised the manuscript accordingly and believe we have addressed the points raised. Below, we provide a detailed, point-by-point response to each of the Reviewers' comments.

Reviewer #1 (Remarks to the Author):

The study by Cunniff et al. reports that KLF5 is highly expressed in pancreatic ductal adenocarcinoma (PDAC) and plays a key role in regulating classical and basal lineage identity. Mechanistically, the authors demonstrate that KLF5 interacts with transcriptional coactivators RUVBL1/2 via its disordered region to modulate cis-regulatory elements in PDAC cells. Importantly, they show that chemical inhibition of RUVBL1/2 suppresses KLF5 function and lineage identity genes. This work is compelling, well-executed, and merits publication with minor revisions to clarify certain points and expand discussions.

Major Points for Clarification/Discussion:

1. KLF5 and H3K27ac Regulation: The authors frequently use H3K27ac ChIP-seq as a readout for cis-regulatory elements. Could they discuss whether KLF5 regulates H3K27ac levels directly (e.g., through TIP60/P400) or indirectly?

> In our study, we used H3K27ac as a marker of global cis-regulatory element activity. This is supported by a substantial body of evidence in the epigenomics field showing that H3K27ac levels correlate with enhancer/cis-regulatory element activity in mammalian cells, a finding that can be traced back to a landmark study published in 2011 (PMID: 21160473). H3K27 acetylation is catalyzed almost entirely by the p300/CBP acetyltransferases in mammalian cells (e.g. PMID: 21131905), with TIP60/P400 not being associated with maintaining this specific mark. TIP60 is more associated with maintaining H2A/H2AZ acetylation. It is important to note that histone gene mutation experiments have shown that H3K27 acetylation is not a functional requirement for enhancer activity (PMID: 35668298) but is instead merely a correlate of enhancer activity (H3K27ac potentially functions redundantly with other histone acetyl marks). For this reason, we did not investigate the mechanism by which KLF5 influences H3K27ac, but an interaction with p300 has been reported previously (Miyamoto et al. 2003; PMID: 14612398). Of note, p300 scored in our marker-based screen (Figure 3D), but it was not detected in our KLF5 pull-down proteomic analysis (Figure 3E). To acknowledge the point raised by the Reviewer, we have added additional text on page 10 describing p300 in our screening data and cite the Miyamoto et al. 2003 study.

Alternatively, could this be further explored by analyzing enhancer RNAs (eRNAs) via SLAM-seq?

> We agree that it is important to validate changes in H3K27ac as a reliable marker of cis element functional changes. Unfortunately, the SLAM-seq library preparation protocol includes an oligo(dT) priming step, such that only newly synthesized RNA with polyadenylation is sequenced. Since eRNAs are usually not polyadenylated, this class of RNAs is absent in the SLAM-seq analysis of our study. Nevertheless, our claim that specific cis-regulatory elements/enhancers are activated by KLF5-RUVBL1/2 is supported by the correlation between changes in H3K27ac levels and changes in newly synthesized mRNAs detected using SLAM-seq, which provides two orthogonal methods showing that perturbations of KLF5 and

RUVBL1/2 suppress the activity of a common set of cis-regulatory elements. Our revised manuscript includes this new analysis.

New Extended Data Figure 3C. Changes in H3K27ac following KLF5 degradation correlate with effects on newly synthesized RNA detected using SLAM-seq. (C) SLAM-seq analysis in AsPC-1 or T3M-4 following 4-hour total treatment with 300nM dTAG^Y-1 or dTAG^Y-1-NEG (control), including 2-hour co-treatment with 500 μ M 4sU. Representative of 2-3 biological replicates. Gene Set Enrichment analysis (GSEA) using gene sets comprised of all HOMER-annotated protein coding genes adjacent to regions of significantly decreased H3K27ac signal following dTAG treatment. See also Supplemental Table 10. NES = normalized enrichment score. FWER = family-wise error rate.

2. KLF5-RUVBL1/2 Axis in Cancer Biology: The interaction between KLF5 and RUVBL1/2 is intriguing. Could the authors speculate whether this axis plays a role in other cancers? For instance, in other malignancies, might RUVBL1/2 cooperate with different lineage-specific transcription factors? Additionally, is this mechanism specific to cancer progression, or could it also have developmental implications?

> We did not fully explore this point in our initial submission. In our revision, we are now including the following results to expand upon the importance of the KLF5-RUVBL1/2 axis in colorectal cancer. We treated T84 human colorectal cancer cells with CB-6644 for 24 hours and performed RNA-seq analysis (New Extended Data Figure 8D). This revealed significant suppression of direct KLF5 target genes, which mirrors effects observed in PDAC models. This suggests that the KLF5-RUVBL1/2 axis is present in additional KLF5⁺ tumor types.

New Extended Data Figure 8D. Inhibition of RUVBL1/2

with CB-6644 suppresses expression of KLF5 target genes in the T84 colorectal cancer cell line. RNA-seq analysis following 24-hour treatment with 750 nM CB-6644 in the T84 colorectal cancer. Fold change was calculated with DESeq2. Gene set enrichment analysis performed on the differentially expressed genes following CB-6644 treatment. NES = normalized enrichment score, FWER = family-wise error rate. See also Supplemental Table 9.

In our original manuscript, we showed that RUVBL1/2 associates with KLF5, MYC, and POU2F3 (a tuft lineage-defining transcription factor) in a AAA+ ATPase-dependent manner. In our revised manuscript, we have expanded our biochemical analysis by showing that RUVBL1/2 can associate with SOX10, a lineage master regulator of melanocyte identity and a lineage dependency in melanoma. Like the association with KLF5/POU2F3/MYC, this interaction is abrogated by inhibiting the ATPase activity of RUVBL1/2 with CB-6644 (New Extended Data Figure 8B). To expand the functional evaluation of these interactions, we performed RNA-seq analysis in a POU2F3⁺ small cell lung cancer and SOX10⁺ melanoma cell lines treated with CB-6644. Notably, in both of these models we observed significant suppression of lineage identity genes (New Extended Data Figure 8K). Together, these findings strengthen our claim of a broader role of RUVBL1/2 in supporting lineage identity in human cancer. These implications are also present in the modified Discussion section of our manuscript on page 19.

Extended Data Figure 8B. RUVBL1/2 associates with MYC, POU2F3, and SOX10 in an ATPase-dependent manner. Western blot analysis following FLAG-TF and HA-RUVBL1/2 co-immunoprecipitation from HEK293T nuclear lysate. HA immunoblotting detects RUVBL1/2 IP. 0.5% input loaded for each sample. FLAG immunoblot is a loading control. Representative of two biological replicates.

New Extended Data Figure 8K. Chemical inhibition of RUVBL1/2 leads to suppression of lineage-specific genes in colorectal cancer, tuft cell-like small cell lung cancer, and in melanoma. RNA-seq analysis following 24-hour treatment with 750 nM CB-6644 in the indicated cell lines. Fold change was calculated with DESeq2. Gene set enrichment analysis performed on the differentially expressed genes following CB-6644 treatment. NES = normalized enrichment score, FWER = family-wise error rate. See also Supplemental Table 9.

RUVBL1/2 interactions with TFs are likely to have relevance in normal developmental processes, though this is clearly speculative and beyond the scope of our study. To acknowledge this important consideration, we now refer to this future direction in the Discussion of our revised manuscript on page 19.

3. Evolutionary and Mechanistic Insights: The analogy between NtrC1 AAA+ ATPase and RUVBL1/2 in transcriptional activation provides valuable evolutionary context. Could the authors elaborate further in the discussion (or even abstract) on the conserved nature of this regulatory cascade?

> NtrC1 is a prokaryote-specific AAA+ ATPase with potential conceptual overlap to the functions of RUVBL1/2 in mammalian systems. NtrC1 has a C-terminal sequence-specific DNA-binding domain that recognizes an inverted motif present at upstream enhancer elements of genes transcribed by G⁵⁴-dependent RNA polymerase (Vidangos et al. 2013; PMID: 23818155). Once bound to these enhancers, NtrC1 loops to the target promoter and uses its AAA+ ATPase to promote conformational changes of G⁵⁴-dependent RNA polymerase that convert it from a closed to an open conformation to drive transcription (Chen et al. 2010; PMID: 21070941). Since RUVBL1/2 lacks a sequence-specific DNA-binding domain, it has likely evolved instead a promiscuous association with disordered activation domains of specific transcription factors, such as KLF5, to facilitate recruitment to RNA polymerase II-dependent genes in human cells. It has been found previously that recombinant RUVBL1/2 can activate human RNA polymerase II under reconstituted *in vitro* conditions (Wang et al. 2022; PMID: 36171202), but it remains unclear whether it drives conformational changes in RNA polymerase II to mediate these effects, in analogy to the NtrC1 paradigm. Structural approaches could be readily applied to RUVBL1/2 to further elucidate its evolutionary relatedness to NtrC1.

The paragraph above has now been added to page 18 of the Discussion of our revised manuscript.

Additional analyses of RUVBL1/2 mass spectrometry data in specific cell types could further strengthen this argument.

> We agree that mass spectrometry analysis of RUVBL1/2 complexes in multiple cell types could further elucidate whether specific features of disordered activation domains encode an ability to bind RUVBL1/2. We have added this future direction to our Discussion on page 19.

Minor Points:

1. Clarification on RUVBL1/2 ATPase Activity: The statement that "ATPase activity of RUVBL1/2 is required for KLF5 binding" could be misinterpreted as affecting KLF5's chromatin binding. It may be clearer to state that "ATPase activity of RUVBL1/2 is required for its association with KLF5."

> We have modified the text accordingly.

2. SLAM-seq Spike-in Controls: Did the authors use spike-in controls for SLAM-seq normalization? If not, this should be noted in the methods or discussed, especially if global transcription changes could affect data interpretation.

> SLAM-seq protocols do not employ a spike-in normalization to account for global transcriptional changes. Instead, a global transcriptional change can be determined via quantification of the overall T>C conversion rate in the entire RNA sample. As shown in new Supplementary Figure 1C-D, when we quantify T>C conversions we observe that CDK9 inhibition with NVP-2 triggers a clear global reduction in RNA synthesis. In contrast, KLF5 degradation and RUVBL1/2 ATPase inhibition with CB-6644 lack a global

reduction of RNA production, but instead produce gene-specific changes in transcription. The methods section of our revised manuscript notes that spike-in controls were not included in the SLAM-seq protocol.

Supplementary Figure 1C-D. Quality control for SLAM-seq. (C-D) (C) Median Conversion rates and (D) Percentage of called reads containing a conversion for each nucleotide combination in the AsPC-1 and T3M-4 SLAM-seq datasets. Each point is one sequencing sample. Median Conversion Rate is plotted as a percentage of all sequenced bases at each position for that sample, corrected for strand. Bars represent the mean \pm SD.

Reviewer #2 (Remarks to the Author):

In this manuscript, Cunniff et al present a comprehensive study combining multiple genomic approaches, including re-analysis of published data, 4sU-labeling RNA-seq, genome-wide CRISPR screening, protein interactome analysis, and in vivo transplantation assays, to identify KLF5 and its interaction with RUVBL1/2 in regulating epithelial feature genes in both classical and basal-like pancreatic ductal adenocarcinoma. The study is data-rich and technically impressive, generating valuable insights into the molecular mechanisms driving cancer cell proliferation. However, the logical flow and clarity of presentation could be improved to better highlight the novel findings and strengthen the overall narrative. Below are my concerns.

Major:

1. The study's logical flow begins with a reanalysis of a previously published CRISPR screen dataset from the same research group. In addition to the already reported Np63, the authors identified KLF5 as another key factor that shows high expression in both basal-like and classical PDAC subtypes. To investigate KLF5's functional role, the authors employed a degradation system targeting KLF5 in cell lines, coupled with 4sU-labeling SLAM-seq analysis. This approach revealed SERPINB5 as one of the common differentially expressed genes downstream of KLF5. The authors then combined additional CRISPR screening with SERPINB5-based FACS sorting, ultimately identifying RUVBL1/2 through subsequent immunoprecipitation experiments.

Within this logical framework, SERPINB5 emerges as what should be a novel and central player connecting KLF5 to RUVBL1/2 in the proposed regulatory pathway. However, despite this apparent importance, the study provides surprisingly limited experimental validation and discussion of SERPINB5's specific role. This relative neglect of SERPINB5 characterization creates a noticeable gap in the otherwise logical progression from KLF5 to RUVBL1/2, making it challenging to fully appreciate how these components functionally interconnect. The manuscript would benefit significantly from more thorough investigation and discussion of SERPINB5's precise mechanistic role in bridging KLF5's transcriptional regulation with the RUVBL1/2 complex.

Specifically:

a) Throughout the manuscript, including the abstract, introduction, and discussion sections, there is remarkably little discussion about the biological function of SERPINB5 in PDAC, nor its functional relationship with the identified KLF5 and RUVBL1/2 components.

> Owing to space limitations, we were not able to fully elaborate on the biological significance of Serpin B5 and its place in pancreatic cancer biology. As explained below, our data are clear that the protein product of the *SERPINB5* gene is not functionally important for the KLF5-RUVBL1/2 transcriptional mechanism described in our study. First, this is based on DepMap analysis, which (in accord with our own findings) shows that *SERPINB5* is not an essential gene in any cancer cell line model, including in PDAC. This is markedly different from KLF5 and RUVBL1/2 which are powerful PDAC dependencies. Second, we are now including in our revised manuscript experiments in which we performed knockout of *SERPINB5* in AsPC-1 cells, followed by RNA-seq analysis (New Extended Data Figure 4G). This revealed that no significant transcriptional changes were detected (aside from down-regulation of *SERPINB5* itself), which is markedly different from significant transcriptional changes observed upon knockout of KLF5 or RUVBL1/2. These two analyses explain why our study did not propose or investigate any function for Serpin B5 in PDAC. Our choice of Serpin B5 as a reporter was made because its expression levels are exquisitely sensitive and specific to KLF5 loss, and its expression is maintained by KLF5 without the assistance of other sequence-specific transcription factors (Extended Data Figure 4H). For these reasons, Serpin B5 was an ideal reporter for identifying functional coactivators of KLF5 in our study.

Published Data Figure Redacted

Figure for Reviewer 2. *SERPINB5* is not essential human PDAC cell lines. Chronos Dependency scores were obtained from the DepMap database of gene essentiality. *SERPINB5* dependency (Chronos score) in each cell line on DepMap is plotted as an individual datapoint. Negative dependency scores denote that *SERPINB5* knockout in that cell line leads to proliferation arrest. White bars represent the mean, and quartiles.

New Extended Data Figure 4G. RNA-seq analysis showing that *SERPINB5* knockout in AsPC-1 cells does lead to significant changes in gene expression. RNA sequencing analysis performed in AsPC-1 cells on day 5 following CRISPR-Cas9 knockout (KO) of *SERPINB5*, or *ROSA26* (control). Two independent sgRNAs were used for each KO. Fold change was calculated by DESeq2. Volcano plots of differentially expressed genes following *SERPINB5* KO. All significantly differentially expressed genes ($p < 0.01$) are labeled. See also Supplemental Table 4.

b) The data presented in Figure 3F reveal an interesting asymmetry: while knockout of KLF5 significantly affects SERPINB5 expression, knockout of SERPINB5 has no apparent effect on KLF5 levels. This observation raises important questions about whether SERPINB5 expression can truly serve as a reliable indicator of KLF5 activity. If SERPINB5 expression doesn't consistently reflect KLF5 status, this potentially undermines the validity of using SERPINB5 antibody-based FACS sorting as the screening method in the genome-wide CRISPR screen shown in Figure 3C. Since the authors used SERPINB5 expression to identify RUVBL1/2 as key interactors, it becomes critically important to determine whether the functional

interaction between KLF5 and RUVBL1/2 actually depends on SERPINB5. To address this question, the authors should perform SERPINB5 knockout experiments to verify whether RUVBL1/2's binding to KLF5 requires SERPINB5's presence.

> As described in detail above, *SERPINB5* is a powerful downstream target gene of the KLF5-RUVBL1/2 transcriptional protein complex. None of the findings in our study implicate the Serpin B5 protein product as having relevance to the transcriptional function of KLF5. 1) *SERPINB5* is not a dependency in PDAC; 2) *SERPINB5* knockout does not change gene expression in PDAC cells; 3) our KLF5 proteomics experiments did not recover Serpin B5 protein as an interacting partner of KLF5; 4) recombinant KLF5 and RUVBL1/2 proteins associate in vitro without the presence of Serpin B5; and 5) ectopically expressed KLF5 associates with RUVBL1/2 in HEK293T, a cell type that lacks *SERPINB5* expression. For these reasons, there is insufficient justification to investigate further a role of Serpin B5 protein as involved in the transcriptional function of KLF5, while there is indeed evidence in the literature that Serpin B5 might have an epigenetic regulatory function in other contexts.

c) While the SLAM-seq data presented in Figures 2C-D and 5I-J (measuring newly synthesized RNA) theoretically provide advantages over total mRNA analysis by distinguishing transcriptional effects from post-transcriptional regulation, the manuscript would benefit from directly comparing these results with using total RNA expression data from the same samples. Specifically, it would be particularly informative to see whether the key findings regarding SERPINB5 hold true in only newly synthesized rather than total RNA analyses, as one would expect the most robust hits (like SERPINB5) to be only identified in new RNA datasets.

To address the Reviewer's comment, our revised manuscript now includes a comparison of newly synthesized and mature RNA within the SLAM-seq datasets (Extended Data Figures 2H and 8J). While *SERPINB5* is downregulated in both the newly transcribed RNA and in the total RNA, the magnitude of reduction is greater in the newly synthesized RNA pool. Examination of other classical versus basal lineage markers reveals similarities and differences between the two RNA pools, which is likely to reflect the variable half-life of specific mature mRNAs: mRNAs with shorter half-lives will tend to change similarly in the newly transcribed and total RNA pools while mRNAs with longer half-lives will decrease disproportionately in the newly transcribed RNA pool.

Figures 2C-D and New Extended Data Figure 2H. Comparison of effects of KLF5 degradation on newly synthesized RNA (top) or on total RNA (bottom). SLAM-seq analysis following 4-hour total treatment with 300nM dTAG^v-1 or dTAG^v-1-NEG (control), including 2-hour co-treatment 500 μ M 4sU. Representative of 2-3 biological replicates. Fold change of gene expression following dTAG^v-1 treatment was calculated with DESeq2. Volcano plots of differentially expressed transcripts following dTAG^v-1 treatment of the indicated cell lines. Select genes within the Moffitt Classical and Basal-like gene sets and *SERPINB5* are labeled. (C-D) Newly synthesized RNA reads or (H) All reads in the sequencing dataset are included in the analysis.

Figures 5I-J and New Extended Data Figures 8J. Comparison of effects of CB-6644 on newly synthesized RNA (top) or on total RNA (bottom). SLAM-seq analysis following 4-hour total treatment with 750 nM dTAG^V-1 or dTAG^V-1-NEG (control), including 2-hour co-treatment 500 μ M 4sU. Representative of 2-3 biological replicates. Fold change of gene expression following CB-6644 treatment was calculated with DESeq2. Volcano plots of differentially expressed transcripts following CB-6644 treatment of the indicated cell lines. Select genes within the Moffit Classical and Basal-like gene sets and *SERPINB5* are labeled. (5I-J) Newly synthesized RNA reads or (8J) All reads in the sequencing dataset are included in the analysis.

Additionally, to properly validate the SLAM-seq methodology, the authors should include quality control metrics in at least the supplementary materials. These should include data on the efficiency of 4sU labeling and chemical conversion, such as the T→C mutation rates and the proportion of newly synthesized RNA in each sample, as well as controls without chemically induced mutation. Such QCs are essential for readers to evaluate the robustness and validity of the SLAM-seq analysis.

We have added a new supplementary figure (Supplementary Figure 1) to our revised manuscript reporting on quality control metrics for our SLAM-seq experiments. This includes a 4sU titration, evaluating effects on cell viability in AsPC-1 and T3M-4, which allowed us to identify non-cytotoxic concentrations of 4sU (panel A). In addition, we performed a global measurement of 4sU incorporation time course via HPLC (panel B). These two controls allowed us to determine optimal 4sU conditions for our cell line models. In panel C, we show the quality control metrics of median T->C conversion rates and percent converted reads from the sequencing data for each of the biological replicates in our study.

Supplementary Figure 1. Quality control for SLAM-seq. (A) CellTiter-Glo analysis in AsPC-1 and T3M-4 8 hours following treatment with 4sU. Luminescence of each 4sU treated sample was normalized to the luminescence of the DMSO treated sample. Two technical replicates are plotted for each cell line. Black line = sigmoidal interpolation \pm 95% confidence interval. Red line = 90% of DMSO. (B) HPLC analysis of 4sU incorporation in AsPC-1 and T3M-4. Points indicate the abundance of identified 4sU nucleosides, relative to U nucleosides. Two technical replicates are plotted for each cell line. Black line = sigmoidal interpolation, Blue line = 1%, Red line = 0.5%. (C-D) (C) Median Conversion rates and (D) Percentage of called reads containing a conversion for each nucleotide combination in the AsPC-1 and T3M-4 SLAM-seq datasets. Each point is one sequencing sample. Median Conversion Rate is plotted as a percentage of all sequenced bases at each position for that sample, corrected for strand. Bars represent the mean \pm SD.

2. The study presents comprehensive datasets that are both data-intensive and scientifically substantial. However, the current visualization of these results in the figures presents significant challenges for reader comprehension. The graphical representations could be substantially improved to enhance clarity and facilitate easier interpretation of the important findings. Several specific issues with the figure presentations should be addressed. For example, multiple figures lack essential labeling information that is crucial for proper interpretation. For example: Figures 2C and 2D: The x-axis label "log2(fold change)" in these figures lacks explanation of what specific comparison is being shown. Figures 2H and 2I similarly lack y-axis labels. Key axis information is also missing in Figures 3B, 3G, 5F and 5K. These labeling deficiencies extend to several supplementary figures as well.

> We thank the Reviewer for bringing this important point to our attention. We have thoroughly reviewed each figure of our study and added additional information to the figure itself or the legend wherever possible to maximize clarity. These modifications were made to Figures 2B-D, 2H-I, 3B, 3G, 5C, 5F, and 5H-K.

Additionally, the relationship between the three heatmaps presented in Figure 4H requires clarification. It remains unclear whether all three heatmaps represent the identical set of 9,274 peaks or if there are differences between them.

> To improve the clarity of Figure 4H, we changed the legend text to, “Heatmaps for KLF5, RUVBL1, and H3K27ac across all 9,274 KLF5 peaks (10 kb window centered on KLF5 peak summits). Rows are ordered by KLF5 signal in the NEG condition, and this ordering is applied to all heatmaps.” We have made similar textual changes to the legends of Figure 5C and Extended Data Figures 3E, 6I, 6K, and 6O to improve clarity as well.

3. In figs. 4D-4F, the IP experiments identify IDR2 as the essential domain for KLF5-RUVBL1/2 interaction. However, Figure 4F shows that the depletion of other domains (DBD and IDR) displays similar interaction patterns to IDR2 deletion. This suggests the possibility that KLF5's function may not rely solely on RUVBL1/2 binding through IDR2, and that additional mechanisms could be involved? The authors should clarify and discuss this observation.

> Figure 4F of our original manuscript is a gene complementation assay showing that deletion of the IDR2, deletion of the entire IDR (IDR1-IDR4), and deletion of the DNA binding domain (DBD) leads to a complete inability of KLF5 to support PDAC cell proliferation. The requirement for the DBD was expected, as it is the critical region of KLF5 required to bind the genome. Figure 4D shows co-IP experiments revealing that deletion of the entire IDR and deletion of IDR2 diminishes the ability of KLF5 to bind to

RUVBL1/2. Moreover, Figure 4E reveals that IDR2 of KLF5 alone (without the DBD) is sufficient to associate with RUVBL1/2, and we show that the DBD deletion binds to RUVBL1/2 in a similar manner to full-length, wild-type KLF5 (Extended Data Figure 5D). Together, these sets of findings suggest that KLF5 binds to RUVBL1/2 through a functionally important IDR2 region within its IDR. Our findings point to IDR2 as an important region of KLF5 involved in its interaction with RUVBL1/2. However, the Reviewer is correct that additional regions of KLF5 could be involved that cooperate with IDR2 in the setting of the full-length KLF5 molecule. We acknowledge this possibility in the revised Discussion of our paper on page 18.

Additionally, why does KLF5 over-expression fail to completely restore function to 100%?

> In our western blot experiments comparing the lentivirally expressed 3xFLAG-KLF5 to endogenous KLF5, we observe consistently that the ectopic 3xFLAG-KLF5 protein is less expressed than the endogenous KLF5. This occurred despite testing multiple lentiviral expression constructs. While this level of expression is clearly sufficient to support PDAC growth upon endogenous KLF5 knockout, it does not quite achieve 100% restoration of function. We have included this Western blot in new Extended Data Figure 5G.

New Extended Data Figure 5G. Western blot analysis of AsPC-1 cells transduced with 3xFLAG-KLF5 cDNA. Protein expression of endogenous and ectopic KLF5 in AsPC-1 PDAC cells. Lane containing 3xFLAG-KLF5 cDNA expression construct is labeled. β-actin is a loading control.

Minor:

Line 69: Remove duplicate "cell lines"

> We have corrected this error.

Fig. 3C: Define "corepressors" and "coactivators"

> Figure 3C legend now includes: "Corepressors and coactivators refer to cofactors that repress or promote KLF5 transcriptional activity, respectively."

Fig. 3H: Clarify CDK1's role

Standardize nomenclature: "RPA3" (Fig. 3H) vs. "RPAP3" (Line 235)

> Figure 3G and Extended Data Figure 4H now include labels indicating that *RPA3* and *CDK1* KO are pan-essential genes involved in DNA replication, included as controls. We included these genes as controls for the RNA-seq experiment to distinguish KLF5-specific transcriptional regulation from non-specific gene expression changes incurred by proliferation arrest. RPAP3 and RPA3 (nomenclature is correct) are in fact two different genes, RPAP3 is a subunit of the R2TP complex. RPA3 is a single stranded DNA-binding protein needed for DNA replication.

Fig. 5G: Explain what CB-644 and dTAG-1 are (inhibitors?)

> The Figure 5 legend now defines CB-6644 (allosteric RUVBL1/2 AAA+ ATPase inhibitor) and dTAG (a bivalent small molecule that bridges the VHL E3 ligase to the FKBP12^{F36V}). In our experiments, KLF5 is tagged with FKBP12^{F36V}, which allows for dTAG-mediated acute degradation via the ubiquitin proteasome system.

Reviewer #3 (Remarks to the Author):

Dear Editor,

I have reviewed the manuscript entitled “KLF5 drives dichotomous lineage programs in pancreatic cancer via AAA+ ATPase coactivators” by Cunniff, et al. This study focuses on the dichotomous role of KLF5 for both the classical and basal-like subtypes of PDAC. Furthermore, the authors find that the AAA+ ATPases RUVBL1 and RUVBL2 are cooperating with KLF5 across both subtypes as coactivators through interaction with the IDP2 of KLF5. This interaction drives the expression of both key classical and basal genes. RUVBL1 and 2 recruitment at these loci is dependent on KLF5, while KLF5 recruitment itself is not dependent on RUVBL1/2 even though H3K27 acetylation is.

This study has considerable strengths, reflecting both the scope of the work and the rigor of its experimental design. The authors comprehensively examine the role of KLF5 in both the classical and basal-like subtypes of PDAC, revealing how a single transcription factor can drive two distinct transcriptional programs. A key and highly novel finding is that the AAA+ ATPases RUVBL1 and RUVBL2 directly interact with KLF5. For all intents and purposes, the same KLF5-RUVBL1/2 complex is shown to regulate the expression of both classical and basal marker genes. In doing so, the work complements and refines models that are driven by distinct master regulators of PDAC subtypes by identifying a shared TF/co-activator unit that both subtypes utilize. The study shows that once KLF5 is bound, RUVBL1/2 are universally required to establish H3K27 acetylation and drive robust transcription in both contexts. Mechanistically, RUVBL1/2 recruitment to enhancer loci is strictly KLF5-dependent, whereas KLF5 binding occurs independently of RUVBL1/2, establishing a clear hierarchy in complex assembly. By uncovering a common dependency on RUVBL1/2 across PDAC subtypes, the study identifies a unifying therapeutic vulnerability with the potential to overcome subtype-specific resistance mechanisms.

While the study has considerable strengths, several limitations and unanswered questions remain that could benefit from further clarification.

While this work is extensive and methodologically sound in many respects, several areas limit the depth and clarity of its conclusions. The central mechanistic question of how KLF5 distinguishes between classical and basal-specific target genes remains unaddressed, which leaves a gap in understanding the true mechanistic basis of how this TF is functioning and selective of specific basal vs classical genes if in reality it is able to select either.

> Our study reveals cis-regulatory element sequence features that distinguish classical- from basal-specific genomic occupancy and gene activation by KLF5 (HNF4A motifs are present at classical-specific KLF5 peaks and p63 motifs are present at basal-specific KLF5 peaks). This provides a sound mechanistic model to account for the context-specific KLF5 functions reported in our study. Deeper mechanistic investigation of this sequence-function relationship is possible, but our focus instead was on defining transcriptional coactivators (RUVBL1/2). We have ongoing follow-up research investigating the sequence-function relationship of classical versus basal-specific KLF5 peaks, but this is beyond the scope of the current work.

Also, relying exclusively on long-established commercial lines may not fully capture the molecular and phenotypic diversity of these subtypes as they occur in patient tumors, which occur more along a continuum.

> An experimental model of cancer (mouse versus human; tissue culture versus organism) should be chosen based on its suitability for addressing the specific biological question being asked. We have gone to great lengths in our study to identify human cancer cell line models that accurately reflect the lineage states of human PDAC observed in clinical specimens (see Figure 1H and Extended Data Figure 1I). Importantly,

we rejected the use of many human PDAC cell line models because they insufficiently reflect the lineage stages observed in the human disease. While our study focuses on PDAC models that possess clear-cut classical (HNF4A+) or basal (p63/KRT5+) categorization, we acknowledge that hybrid classical/basal states are likely to exist in human tumors and the role of KLF5 in such cells remains unknown. We acknowledge this limitation of our study in our revised Discussion on page 16.

The study frames its results as a discrepancy with prior reports citing KLF5 loss in high-grade PDAC or during EMT, yet several other studies have documented high KLF5 expression in PDAC, suggesting the current findings are not necessarily in conflict and could be more accurately positioned within the broader literature.

> Diaferia et al. 2016 [PMID: 26769127] reported that KLF5 expression was present in classical PDAC but absent in what they termed ‘high-grade’ PDAC. This study did not explicitly examine basal-squamous PDAC. David et al. 2018 [PMID:26898331] subsequently reported that KLF5 expression became silenced during EMT in PDAC models, which is in accord with the findings from Diaferia et al. Other studies around this time also report high KLF5 expression in PDAC, but did not link this expression to lineage subtypes of this disease (e.g. He et al. 2018; PMID: 29248441). Finally, a ‘high-grade’ histological appearance of PDAC has been shown to correlate with basal-squamous identity (Puleo et al. 2019; PMID: 30165049). Taken together, this body of evidence would suggest that KLF5 would be absent in basal-like PDAC, although this was never formally demonstrated in a prior study. An important conclusion from our work, with substantial experimental support, is that KLF5 is expressed in basal-like PDAC and maintains this identity. The Reviewer is correct that the term ‘discrepancy’ is a possible overstatement, and therefore we have carefully positioned and reconciled our observations in our Results (page 6) and Discussion (page 16) of our revised manuscript.

The results tend to stick to their own lane, in other words, effects are assessed only within each subtype such that basal lines are evaluated for only basal-associated genes, and classical lines for classical-associated genes. Even if the alternate program is minimally or not expressed at baseline, the study should test whether perturbing KLF5 or RUVBL1/2 produces any measurable impact on the other program. Such analyses would clarify whether the proposed shared co-activator module is truly insulated by cell state or has the potential to influence the alternate transcriptional program.

> While the AsPC-1 transcriptome is dominated by expression of classical identity genes, there is indeed a low-level expression of a subset of basal identity genes (e.g. *FAM83A*). Using our acute degradation SLAM-seq datasets, we found that KLF5 is in fact needed to maintain both classical and basal identity genes in this model. We did the converse analysis in T3M-4 cells (a basal model), which revealed a role for KLF5 in maintaining the low-level expression of select classical identity genes in this system. Overall, these findings support our claim of a dichotomous lineage function of KLF5 in maintaining classical or basal identity gene expression, in cooperation with other subtype-defining transcription factors. These results are included in our revised manuscript as Extended Data Figures 2I-J.

New Extended Data Figures I-J. KLF5 maintains expression of Classical Identity genes in T3M-4 cells and maintains expression of Basal identity genes in AsPC-1 cells. SLAM-seq analysis in T3M-4 and AsPC-1 following 4-hour total treatment with 300nM dTAG^v-1 or dTAG^v-1-NEG (control), including 2-hour co-treatment with 500 μ M 4sU. n=2-3. Fold change of T⁻C converted transcripts following dTAG^v-1 treatment were calculated with DESeq2. Gene set enrichment analysis performed on differentially expressed transcripts following KLF5 degradation. NES = Normalized Enrichment score. FWER = Family-wise Error Rate.

The observation that common KLF5 peaks are predominantly promoter-proximal, while subtype-specific peaks are enriched in intergenic or intronic regions, lacks deeper exploration of the functional implications of this distribution (e.g. enhancers/super-enhancers/chromatin architecture).

> Numerous studies indicate that intergenic and intronic binding sites of transcription factors, particularly when enriched for H3K27ac, correspond to enhancers and tend to regulate cell type-specific gene expression. Therefore, an important conclusion from our study is that classical and basal subtype-specific KLF5 peaks tend to be distal enhancers, which we now state on Page 9 of our revised manuscript. Using our genome-wide computational analyses, we have shown that HNF4A and p63 motifs are a strong sequence correlate of these distal enhancers, which tend to be less enriched at promoters. This provides a mechanistic model to account for this observation.

Maybe the authors can clarify why in their sgKLF5 RNA-seq raw data that KLF5 itself is not significantly down with other KLFs more significantly impacted in each of the cell lines.

> Nonsense-mediated decay (NMD) will not lower mRNA levels if the premature stop codons occur in the final coding exon. The KLF5 sgRNAs that we used for this analysis target the C-terminal zinc finger domains (to ensure that frameshift and in-frame deletions will both cause loss-of-function), which is encoded by the final exon. Therefore, we did not expect to find a significant decrease in the KLF5 mRNA levels from CRISPR induced indels in this exonic location, as NMD cannot occur.

In vivo, KLF5 expression was restored in endpoint tumor samples despite evidence of effective knockout at earlier time points. This raises questions about the cellular source of the recovered signal that are not examined, such as stromal contamination in bulk tissue analysis, repopulation by unedited tumor cells, or selection for resistant/reversion variants. These issues are important to address, as the current data are used to support the conclusion that KLF5 is a “genetic dependency” in both classical and basal-like PDAC models.

> In our revised manuscript, we are including new genomic analyses to support that endpoint KLF5 knockout tumors have regained KLF5 function (Extended Data Figures 3M-P). Using RT-qPCR of the GFP

transgene linked to the lentiviral sgRNA expression vector, we find that 6 out of 10 KLF5 KO tumors have nearly undetectable GFP expression, indicating a loss of sgRNA⁺ cells from these tumors. Next, we performed Sanger sequencing of the genomic DNA site targeted by the KLF5 sgRNA. This revealed that the same 6 tumors that lacked GFP/sgRNA expression had a wild-type coding sequence of KLF5, as expected. For the 4 tumors that retained GFP expression, we found instead that these tumors had selected for in-frame insertions or deletions in the KLF5 coding sequence. These in-frame alleles are expected to retain functionality. Finally, we performed RT-qPCR using mouse and human-specific primers detecting KLF5 to distinguish human tumor expression from mouse stroma expression. These revealed that the human *KLF5* expression was >20-fold higher in levels than the mouse *Klf5* gene, supporting that our original western blot of *KLF5* KO tumors was in fact measuring tumor KLF5 expression and not *Klf5* expression in stroma. Taken together, these results further strengthen our conclusion that KLF5 is a genetic dependency in PDAC tumor models.

New Extended Data Figures 3M-P. Additional genomic analysis of KLF5 KO xenografts from AsPC-1 and T3M-4 cells. (M, P) RT-qPCR analysis of (M) *GFP* transgene (coupled to sgRNAs in the lentiviral expression vector) and (P) *KLF5* in resected orthotopic AsPC-1 and T3M-4 xenografts. Normalized expression values are calculated as $2^{-\Delta Ct}$ normalized to the expression of *ACTB* in the same tumor sample. Each measurement is plotted, and bars represent the geometric mean of four measurements \pm SD. (N,O) Sanger sequencing of *sgKLF5* cut site from Genomic DNA from resected AsPC-1 and T3M-4 xenografts. *sgRNA* recognition sequence is labeled with a purple box. Canonical *KLF5* DNA sequence is annotated below each track. (O) Annotation of the identified *KLF5* DNA mutations in each tumor sample. Individual

tumor IDs are indicated to the left of the table. + = insertion, - = deletion, WT = canonical sequence detected. (P) Human *KLF5* and mouse *Klf5* expression values were both normalized to human *ACTB*.

For the mass spectrometry immunoprecipitation, it is unclear which PDAC lysate was used, and because recombinant KLF5 was employed, interactions dependent on endogenous post-translational modifications may have been missed.

> We have modified the text in the Figure 3 legend to denote that both the CRISPR screen and mass spectrometry (MS) analysis were performed using AsPC-1 cells.

We agree with the reviewer that our experimental design has limitations. Purified recombinant KLF5 protein was used as bait for the pulldown assay to maximize sensitivity of the proteomic pull-down assay, given the weak interaction strength between transcription factors and their cofactors (Wright and Dyson, 2015; PMID: 25531225). We acknowledge that a limitation of this approach is in missing potential PTM-mediated interactions, and our revised Discussion on page 17 acknowledges this point.

Although RUVBL1/2 peaks show some subtype specificity, their overall genomic distribution is skewed toward promoters (while the more subtype specific KLF5 peaks skew toward intergenic/intronic), and the BxPC-3 basal-like ChIP-seq dataset appears less convincing for the basal-specific peaks than the classical AsPC-1 dataset for its classical-specific peaks. Given this concern, the authors' conclusion "As observed with KLF5, we found that a significant proportion of RUVBL1/2 peaks were specific to classical or basal-like PDAC models" is not fully supported.

> Our revised manuscript includes additional analyses to better highlight classical and basal-specific RUVBL1/2 peaks. Extended Data Figure 6C uses a heatmap representation to more clearly extract out subtype-specific RUVBL1/2 peaks without limiting the analysis to KLF5 peaks. In addition, we have replaced the original Extended Data Figure 4F (heatmap) with Extended Data Figure 6G, which instead uses a metaplot to better highlight the variable occupancy of RUVBL1/2 at KLF5-occupied, subtype-specific binding sites. Finally, we also include browser tracks of representative loci to demonstrate the binding profiles at specific genes.

New Extended Data Figure 6C. Heatmaps of cell line-specific and common RUVBL1 peaks detected by ChIP-seq. (C) RUVBL1 ChIP-seq in AsPC-1 and BxPC-3. Each row is a 10Kb genomic region centered on a RUVBL1 peak. Each column is one replicate in one cell line, labeled above. AsPC-1-specific, BxPC-3-specific, and Common RUVBL1 peaks were defined by a bedtools intersect analysis of RUVBL1 peaks (MACS2 $q < 0.01$). See also Supplemental Table 6.

New Extended Data Figure 6E. Representative genomic tracks of cell type-specific RUVBL1 peaks. (E) ChIP-seq tracks showing KLF5, RUVBL1, and H3K27ac enrichment at the indicated loci, visualized in the UCSC genome browser. Matched, scaled track heights are listed on the right.

New Extended Data Figure 6G. Metaprofiles depicting RUVBL1 occupancy at cell type-specific sites and common sites. (G) RUVBL1 ChIP-seq in AsPC-1 and BxPC-3. Classical, Basal-like, or Common KLF5 peak are defined in Figure 2E. See also Supplemental Table 6. Metagene plots show the average signal for RUVBL1 across all peaks in each peak set, separated by cell line.

Functional assessment of RUVBL1/2 would benefit from chromatin accessibility data (ATAC-seq), which would better capture their direct role in transcriptional regulation than H3K27ac ChIP-seq alone.

> In our experience and that of others, a significant subset of accessible enhancers identified by ATAC-seq will not be actively regulating transcription, but instead will be poised for activation by another stimulus (PMID: 34874265). The original identification of H3K27ac as an enhancer mark, noted that it can distinguish poised enhancers from active enhancers (PMIDs: 21160473 and 21106759). The goal of our study was not to investigate poised enhancers or chromatin opening, but instead to demonstrate which enhancers are activated by KLF5 and RUVBL1/2. Therefore, we chose H3K27ac as our epigenomic mark for annotating enhancer activity genome wide. Importantly, there is no evidence that RUVBL1/2 uses its ATPase for chromatin remodeling and our work clearly indicates an INO80 complex-independent function in PDAC. For these reasons, we do not see a strong justification to replicate our epigenomic analysis with ATAC-seq analysis.

Minor comments:

The introduction is quite lengthy, including tangential information (e.g., basal-like PDAC hypersensitivity to KRAS inhibitors) that is not directly relevant to the current work and could be more focused.

> We have removed this sentence.

Line 51 “progrosis” should presumably be “prognosis”

> We have made this correction.

Antibody information was provided on the reporting summary but would be beneficial to have within the methods or as a supplementary table for the various uses (ChIP-seq versus WB, etc).

> We added antibody information as Supplementary Table 13.

Reviewer #4 (Remarks to the Author):

As requested by the editor, this review focuses on the execution, presentation, and interpretation of the pooled CRISPR screens in the manuscript "KLF5 drives dichotomous lineage programs in pancreatic cancer via AAA+ ATPase coactivators".

The authors performed both genome-wide KO and TF-focused CRISPRi screens in AsPC-1 PDAC cells using an endogenous SERPINB5 (Serpins B5) reporter, binned by flow cytometry into LOW, BULK, and HIGH reporter populations. This approach is appropriate for identifying regulators of the SERPINB5 locus and its upstream transcriptional network.

The genome-wide screen employs the Brunello library, targeting most genes with 4 sgRNAs, which is standard and sufficient for robust identification, whereas the TF screen uses a CRISPRi approach. The reporter screen recovers expected hits such as SERPINB5 (the reporter itself) and KLF5 as the most significantly depleted genes in the Serpin B5-high bin. This demonstrates screen functionality, sensitivity, and specificity. Screens maintain very high coverage for each sgRNA and are sequenced sufficiently deeply. Overall, screen execution, presentation, and interpretation are meticulous, and the screen quality is good. Raw count tables for each sgRNA across the three groups are provided and easily mapped to gene and group assignments. Three points:

1) The data has several gene names converted to calendar dates, a common error introduced by Excel. This is due to Brunello annotation using outdated HUGO gene IDs, which were updated to avoid this issue. This should be fixed (MARCH1-MARCH11, now called MARCHF1-MARCHF11, SEPT1-12 and SEPT14, now called SEPTINI-12 and SEPTINI14, respectively, and DECI, now DELEC1. Please double check that these gene IDs are correct.

> We have corrected the gene names (MARCHF1-11, MTARC1-2, SEPTIN1-12, 14, and DELEC1) in the datasets uploaded to the GEO Database, as well as in the Supplementary Tables for the CRISPR screen, RNA-seq, and SLAM-seq to reflect the updated annotations.

2) The TF screen sample "BULK" is strange – "High" and "Low" populations correlate well, but the "BULK" population is uncharacteristically different from both. This indicates to me that something is not quite right with the "Bulk" sample. The authors should look into this again to exclude any errors, and if the poor correlation is indeed in the data, mention this in the text. It would then be best to mainly rely on the "high" to "low" comparison for hit calling. To clarify - this is unlikely to affect any of the conclusions and the identity of the hits, however, should be discussed.

> Upon re-examination, we identified a sample barcoding error during library preparation that affected sequencing of the "High" population. To resolve this issue, we re-barcoded and re-sequenced all three libraries for the TF screen and have uploaded the re-sequenced raw and processed data to GEO and updated the original datasets (now named "CRISPR_TF_SERPINB5_JustLowBulk..."). We also updated Extended Data Figure 4H and the data in Supplementary Table 1 to reflect the corrected "Low" vs. "High" comparison, as recommended. The correlation plots for each comparison are provided below.

Figure for Reviewer. sgRNA abundance comparisons in low, high, and bulk populations. sgRNA enrichment in the “High,” “Low,” and “Bulk” populations in the Transcription-factor focused (top) and Genome-wide (bottom) Serpin B5 reporter CRISPR Screen. sgRNAs are depicted as dots. sgRNAs targeting *KLF5*, *SERPINB5*, or negative controls are labeled.

The correlations are now internally consistent across the different comparisons. We are grateful to this Reviewer for identifying this error.

3) *More explicit plots or tables showing sgRNA-level variability and consistency for key hits (e.g., KLF5, SERPINB5, RUVBL1/2) would be helpful and demonstrate results more clearly.*

> We added three new plots to Supplementary Figure 2 showing sgRNA-level variability and consistency for *KLF5*, *SERPINB5*, *RUVBL1*, and *RUVBL2*.

New Supplementary Figures 2C-E. Additional controls for marker-based CRISPR screens. (C-E) sgRNA enrichment in the “High” and “Low” populations in the (C) Transcription-factor focused and (D-E) Genome-wide Serpin B5 reporter CRISPR Screen. sgRNAs are depicted as dots. sgRNAs targeting *KLF5*, *SERPINB5*, or negative controls are labeled. (E) Dashed line = mean. Negative values indicate enrichment in the “low” population.

In summary, the screens are executed, presented and interpreted correctly, but above issues should be addressed prior to publication.

Minor point:

The method section contains a reference to a genome-wide CRISPRi library; however this was apparently not used? Remove if not needed. Also, throughout the manuscript it would be good to refer to “CRISPR-KO” screen or “CRISPRi screen”, to more clearly distinguish the two experiment types.

We previously performed genome-wide reporter-based CRISPRi screens in three basal-like PDAC cell lines (BxPC-3, KLM-1, T3M-4). In Figure 1A, we specifically analyzed the enrichment of sgRNAs targeting transcription factors in these existing screening. To clarify this, we revised the legend to read: “(A) Genome-wide KRT5 reporter CRISPRi screen results in three independent cell lines (KLM-1, T3M-4 and BxPC-3). Average beta scores of 1,599 transcription factors are shown.”

Because these screens were performed and analyzed in our prior study, we removed the genome-wide CRISPRi library reference from our methods section. This section now only refers to the TF-focused and genome-wide (Brunello) CRISPR-KO libraries.